# Functional convergence of biosphere–atmosphere interactions in response to meteorological conditions

Christopher Krich[1,2], Mirco Migliavacca[1], Diego G. Miralles[2], Guido Kraemer[1,3], Tarek S. El-Madany[1], Markus Reichstein[1], Jakob Runge[4], and Miguel D. Mahecha[3,5]

[1]Max Planck Institute for Biogeochemistry, 07745 Jena,Germany
[2]Hydro-Climate Extremes Lab (H-CEL), Faculty of Bioscience Engineering, Ghent University, Ghent, Belgium
[3]Remote Sensing Centre for Earth System Research, Leipzig University, 04103, Leipzig, Germany
[4]German Aerospace Center, Institute of Data Science, 07745, Jena, Germany
[5]Remote Sensing Centre for Earth System Research, Helmholtz Centre for Environmental Research UFZ, 04318 Leipzig, Germany

**Correspondence:** Christopher Krich (ckrich@bgc-jena.mpg.de)

**Abstract.** Understanding the dependencies of the terrestrial carbon and water cycle with meteorological conditions is a prerequisite to anticipate their behaviour under climate change conditions. However, terrestrial ecosystems and the atmosphere interact via a multitude of variables across temporal and spatial scales. Additionally these interactions might differ among vegetation types or climatic regions. Today, novel algorithms aim to disentangle the causal structure behind such interactions from empirical data. The estimated causal structures can be interpreted as networks, where nodes represent relevant meteorological variables or land-surface fluxes, and the links the dependencies among them (possibly including time-lags and link strength). Here we derived causal networks for different seasons at 119 eddy-covariance flux tower observations in the FLUXNET network. We show that the networks of biosphere–atmosphere interactions are strongly shaped by meteorological conditions. For example, we find that temperate and high latitude ecosystems during peak productivity exhibit very similar biosphere–atmosphere interaction networks as tropical forests. In times of anomalous conditions like droughts though, both ecosystems behave more like typical Mediterranean ecosystems during their dry season. Our results demonstrate that ecosystems from different climate zones or vegetation types have similar biosphere–atmosphere interactions if their meteorological conditions are similar. We anticipate our analysis to foster the use of network approaches as they allow a more comprehensive understanding of the state of ecosystem functioning. Long term or even irreversible changes in network structure are rare and thus can be indicators of fundamental functional ecosystem shifts.

## 1 Introduction

Terrestrial ecosystems and the atmosphere constantly exchange energy, matter, and momentum (Bonan, 2015). These interactions result in biosphere–atmosphere fluxes (in particular carbon, water, and energy fluxes) that are shaped by a variety of climatic conditions and states of the terrestrial biosphere (McPherson, 2007). Understanding how biosphere–atmosphere fluxes interact and how they causally depend on the short-term meteorological and long-term climate conditions is crucial for building predictive terrestrial biosphere models (Detto et al., 2012; Green et al., 2017). However, the exact causal structure of dependen-

cies between surface and atmosphere variables is still subject to unknowns (Baldocchi et al., 2016; Miralles et al., 2019). For example, we still do not understand well under which conditions certain climate extremes turn ecosystems into carbon sources or sinks (Sippel et al., 2017; Flach et al., 2018; von Buttlar et al., 2018). One reason for our incomplete understanding is that the causal dependencies underlying biosphere–atmosphere interactions might vary among ecosystems depending on vegetation structure and its long-term adaptation to climatic conditions.

Conducting a comparative study across ecosystems, focusing on their interactions with the atmosphere has two requirements: Firstly, we need standardised data encoding biosphere fluxes and meteorological conditions. Secondly, an analytical tool is needed that extracts an interaction structure from these data empirically. The latter requires handling of multivariate processes and estimating dependencies beyond correlations. The first requirement is best met by the FLUXNET database (Baldocchi, 2014), a collection of global long-term observation of biosphere–atmosphere fluxes measured via the eddy covariance method (Aubinet et al., 2012). The spatial distribution of FLUXNET sites is biased to European and North American sites, yet it still covers most climate zones and vegetation types ranging from boreal steppe to tropical rainforests surprisingly well Reichstein et al. (2014). Further, the data is processed homogeneously across sites. The second requirement is addressed by causal inference. Various methods exist today (see Runge et al., 2019a, for a recent overview), some of which have been applied already in the biogeosciences (Ruddell and Kumar, 2009; Detto et al., 2012; Green et al., 2017; Papagiannopoulou et al., 2017; Shadaydeh et al., 2019; Claessen et al., 2019). One of that group is PCMCI (Runge et al., 2019b), a causal graph discovery algorithm based on a combination of the PC algorithm (named after its inventors Peter and Clark (Spirtes and Glymour, 1991)) and the Momentary Conditional Independence (MCI) test (Runge et al., 2019b). By applying such tests, it becomes possible to account for common drivers and mediators which can cause two variables to correlate even though, no direct causal link exists between them. Then MCI partial correlations estimated by PCMCI yield a better interpretation of the strength of a causal mechanism than the common Pearson correlation. Krich et al. (2020) tested PCMCI regarding its suitability for interpreting eddy covariance data. The method proved to be consistent despite the data's inherent noisy character and was capable to extract well interpretable interaction structures. A causal interpretation of specific links, though, has to take into account potentially unmet assumptions.

In this study, we investigate multivariate time series from FLUXNET tower data to understand how networks of biosphere–atmosphere interactions vary across vegetation types and climate zones. The rationale is as follows: If biosphere–atmosphere interactions varied significantly across climate gradients or between vegetation types, this could indicate, for example, that ecosystem responses to climatic extremes could differ significantly and would require terrestrial biosphere models to account for them differently. If, however, the opposite applies and ecosystems of the Earth exhibit similar biosphere–atmosphere interaction types, then common principles can be identified that can serve as empirical reference for global vegetation models. We hypothesise first that the accessible states of biosphere–atmosphere interactions are limited and can be characterised by few functional states despite the complexity and differences among ecosystems. Second, attributing to an ecosystems adaptation, we further hypothesise that specific ecosystem can only access a limited fraction of the functional states.

The study is designed as follows: Firstly, we perform causal discovery by PCMCI at each eddy covariance site and seasons. Secondly, we solely investigate the resulting interaction networks and visualise them in a low-dimensional space. We then

interpret the low-dimensional space of biosphere–atmosphere interactions and investigate seasonal cycles, characteristic states, the role of vegetation types and finally discuss the potential role of adaptation to the underlying climate space.

## 2 Data and Methods

### 2.1 Eddy-covariance observations

We used eddy covariance data from the FLUXNET database (Baldocchi et al., 2001) aggregated to daily time resolution. To maximise the available ecosystems and time series length, we took the union of the LaThuile Fair use (Baldocchi, 2008) and FLUXNET2015 Tier 1 (Pastorello et al., 2020) datasets (Nelson et al., 2020) with at least 5 years of measurement. If a site year was available in both datasets we selected the one from FLUXNET2015. A detailed list of used sites and years is given in table A1. The final dataset contains 119 sites from the major plant functional types and covers the major Koeppen-Geiger climate classes, i.e. tropical to polar climate zones. The majority of sites belong to evergreen needleleaf forests, grasslands and deciduous broadleaf forests. The dominant climate classes are continental, temperate and dry climates. The dataset's variables, including meteorological and eddy covariance measurements, were quality checked, filtered, gap-filled, and partitioned with standard tools (Papale et al., 2006; Pastorello et al., 2020) and provided with per-variable quality flags. We extracted following variables, comparable between the two dataset, and their corresponding quality controls (if available): shortwave downward radiation (or global radiation, $R_g$), air temperature (T), net ecosystem exchange (NEE) (inverted, so that positive values signify carbon uptake into the biosphere), vapour pressure deficit (VPD), sensible heat- (H), latent heat flux (LE), gross primary productivity (GPP), precipitation (P) and soil water content (SWC, measured at the shallowest sensor). Within the FLUXNET2015 dataset these variables are named as: "SW_IN_F_MDS", "TA_F_MDS", "NEE_VUT_USTAR50", "VPD_F_MDS", "H_F_MDS", "LE_F_MDS", "GPP_NT_VUT_USTAR50", "P", "SWC_F_MDS_1", respectively. Correspondingly for the LaThuile dataset:"Rg_f", "Tair_f", "NEE_f", "VPD_f", "LE_f", "H_f", "GPP_f", "precip", "SWC1_f", respectively. GPP is calculated via the commonly used night time flux partitioning (Reichstein et al., 2005). Here GPP is the difference between ecosystem respiration and NEE. The latter is estimated via a model which is parameterised using night time values of NEE.

### 2.2 PCMCI

To analyse biosphere–atmosphere interactions, we estimated network structures using the causal network discovery algorithm PCMCI. PCMCI is tailored to estimate time-lagged dependencies from potentially high-dimensional and autocorrelated multivariate time series. Dependencies can be interpreted causally under certain assumptions. The algorithm is explained from a biogeoscience viewpoint in Krich et al. (2020). A comprehensive description from theoretical assumptions to numerical experiments is given in Runge et al. (2019b). As a brief summary, PCMCI efficiently conducts conditional independence tests among variables to reconstruct a dependency network. While PCMCI can also be combined with nonlinear tests, here we estimate conditional independence using partial correlation (ParCorr), implying that we only consider linear dependencies. Partial

correlation between two variables $X$ and $Y$ given a variable set $Z$ is defined as the correlation between the residuals of $X$ and $Y$ after regressing out the (potentially multivariate) conditions $Z$. The conditions $Z$ can consist of lagged third variables or time-lags of $X$ and $Y$.

PCMCI has two phases. In the first phase, the 'condition selection', a superset of lagged parents (up to some maximum time lag $\tau_{max}$) of each variable, $X_t^j$, is estimated based on a fast variant of the PC algorithm (Spirtes and Glymour, 1991). A parent of $X_t^j$ is any lagged variable, $X_{t-\tau}^i$, that is directly influencing $X_t^j$. This can be the own past, $i = j, \tau > 0$ or other variables, $i \neq j, \tau > 0$. A pseudo-code of this procedure is given in the supplementary materials of Runge et al. (2019b). In the second phase, 'momentary conditional independence' (MCI) is estimated among all pairs of contemporaneous and lagged variables $(X_{t-\tau}^i, X_t^j)$ for $\tau \geq 0$. The MCI test removes the influence of the lagged drivers (obtained in the first phase) using ParCorr and yields link strengths and p-values (based on a two-sided t-test). The link strength is here given by the MCI partial correlation . In short, the MCI value gives an estimate of dependence between two time series, one potentially lagged, with the influence of other lagged drivers including autocorrelation removed, yielding a better interpretation of the strength of a causal mechanism than the common Pearson correlation. For a more detailed discussion of the interpretation, see Runge et al. (2019b). As a particular partial correlation, the MCI value is independent of the variables' mean value and is normalised in [-1, 1] and can, hence, be compared between variable pairs with different units of measurement. Lagged links are directed forward in time. Contemporaneous dependencies are left undirected as no time information reveals the direction of influence unless they are defined as unidirectional by the user (pcmci parameter selected_links, see table B1). A causal interpretation of links rests on the standard assumptions of causal discovery. Here we assume time order, the causal Markov condition, faithfulness, causal sufficiency, causal stationarity, and no contemporaneous causal effects. The use of ParCorr additionally requires stationarity in the mean and variance and linear dependencies (Runge et al., 2019b). In particular, a statistical independence (here at a 0.1 two-sided significance level) between a pair of variables conditional on the other lagged variables is interpreted as the absence of a causal link (Faithfulness condition). On the other hand, a causal interpretation of the estimated links is here to be understood only with respect to the variables included in the analysis. The dependence structure among variables can finally be visualised by weighted networks with the nodes representing the variables and the links significant dependencies with its strengths given by the MCI partial correlation.

## 2.3 Network Estimation

Dependencies are estimated using PCMCI among the variables $R_g$, T, NEE, VPD, H and LE using time lags ranging from zero to five. As was already discussed by Krich et al. (2020), eddy covariance data and the choice of our variable set do not fully fulfil all assumptions of PCMCI. Causal sufficiency and no contemporaneous links are obviously not fulfilled which can lead to spurious links. Yet, in the present context we aim to compare networks and a causal interpretation of each link is not the focus. We further can not rule out non-linear dependencies. In case they have a strong linear part, we nevertheless can detect them. Based on findings in Krich et al. (2020), we subtracted a smoothed seasonal mean from each variable to remove the common driver influence of the seasonal cycle that would yield spurious dependencies. The seasonal mean was smoothed by setting the high frequency components ($> 20 \, \text{days}^-$) of its Fourier transform to zero. This decreases the detection of false links

while leaving the detection of true links largely unaffected. We estimated networks in sliding windows of three months, taking the centre month as the time index of each network. The sliding windows help to capture the temporal evolution of biosphere–atmosphere interactions and provide enough data points for the network estimation via PCMCI. Additionally, we improve stationarity of the data further and address the requirement of causal stationarity, i.e. a causal link persists throughout the time period of network estimation. Further we set $R_g$ as a potential driver of the system (by excluding its parents from the PCMCI parameter 'selected_links', see table B1). We acknowledge the possibility of $R_g$ being influenced by other variables, e.g. via transpiration and subsequent cloud formation. Yet, on the ecosystem scale we work with, we presume this effect to be rather small and likely dominated by lateral transport. Besides these possibilities, setting $R_g$ as driver can account for remaining non stationarities (Runge, 2018). The analysis was performed also without this setting, i.e. allowing influences of other variables on $R_g$. The conclusions we draw are not affected (cf. Fig. D1). Missing data was flagged as such and is ignored by PCMCI. To avoid effects on the network structure from gap-filling we used the following quality flag thresholds. A daily datapoint is not used if its quality flag is below 0.6 (i.e. more than 60% of measured and good quality gap filled data). In case more than 25% of datapoints of the three month window are flagged as bad quality, the time window is removed from the analysis. To analyse the factors influencing network structure, we consider the mean values over the respective time period of the variables included in the network calculation, and additionally those of GPP, P and SWC. GPP, P and SWC were not included in the network calculation because certain characteristics can impinge on network estimation. GPP is derived using NEE and T. Any of the links GPP–T and GPP–NEE thus could be due to its processing rather than an actual dependence. P, on the other hand, typically yields non intuitive results due to its binary character (precipitation of a certain amount - zero precipitation), while its effects occur more smoothly (e.g. increase in transpiration or respiration) and its strong deviation from a normal distribution. Further, it can happen that over the time period of network estimation no precipitation occurs rendering such periods not analysable. The issue with SWC is its lower availability and for those sites that have such measurements it might be applied at differing depth. The depth that is mostly present is at shallow depths of 5 or 10 cm. The upper soil layer, however, dries out quickly and can explain only little of the latent heat flux.

## 2.4 Dimensionality Reduction

For the dimensionality reduction, we tested principal component analysis (PCA; Pearson, 1901), t-distributed stochastic neighbour embedding (t-SNE; Maaten and Hinton, 2008), and uniform manifold approximation and projection (UMAP; McInnes et al., 2018). PCA is the standard method for dimensionality reduction, it is commonly used, linear, fast, and easily interpretable regarding the meaning of its axes (the principal components). A PCA embedding typically fails to reveal complex clusterings, because it maintains large scale gradients but often produces embeddings in which far away points appear very close in the embedding. In contrast t-SNE aims to preserve local neighbourhoods. Therefore it calculates first similarity scores for each point pair using euclidean distances and Gaussian distributions. Subsequently it randomly projects the data onto the lower dimensional space and attempts to rearrange points in a way that the previously determined similarities are obtained. To assess the similarities in the low dimensional space, however, it uses a Student-t distribution. This helps to separate points which are also originally separated. This procedure makes t-SNE very good at visualising clusters in the data and non-linear

relationships. Drawbacks are the difficult interpretability of the embedding axes due to the non-linear nature and its fairly long computation time for large datasets. Further, distances between far separated points and those belonging to different clusters in the embedding space are not (necessarily) comparable to the original distances. This is as t-SNE does not preserve both the global and local structure at the same time, which is attempted by UMAP. UMAP was developed as an improvement of t-SNE regarding structure preservation and results also in a shorter run time especially for higher dimensions. A comparison of t-SNE and UMAP is given in appendix C in McInnes et al. (2018). According to Kobak and Linderman (2019), the global structure preservation of UMAP is not an inherent characteristic of the method itself but rather stems from the choosen initialization.

As we are dealing with an unsupervised method there is no obvious measure to assess the quality of an embedding, as each method optimises a different error function. A measure commonly used for the comparison and characterisation of dimensionality methods is the agreement between $K$-ary neighborhoods (the $K$ nearest points to an observation) in the high dimensional and low dimensional space. The measure $R_{\mathrm{NX}}(K)$ (Lee et al., 2015) gives a measure of the improvement of the embedding of $K$-ary neighborhoods over random embeddings. For an embedding with random coordinates we obtain $R_{\mathrm{NX}}(K) \approx 0$ and if the $K$-ary neighborhoods are perfectly preserved we obtain $R_{\mathrm{NX}}(K) = 1$. As this measure depends on the neighborhood size, $K$, we can draw a curve over $K$ that characterizes if the method is better at maintaining global or local neighborhoods. The area under the $R_{\mathrm{NX}}(K)$ curve gives an idea of the overall quality of the embedding. An intercomparison of the three dimensionality reduction methods using this measure shows t-SNE to perform best (see Fig. A1, B1, C1).

## 2.5 Distance Correlation

Distance correlation (Székely et al., 2007) is a non linear measure to quantify the dependence between two vectors. It has been used successfully to assess the influence of variables on the low dimensional embedding (Kraemer et al., 2020b). Székely et al. (2007) details its empirical definition for a sample $(\mathbf{X}, \mathbf{Y}) = \{(X_k, Y_k) : k = 1, ..., n\}$ with $\mathbf{X} \in \mathbb{R}^p$ and $\mathbf{Y} \in \mathbb{R}^q$ as follows:

$$
\mathcal{R}_n^2(\mathbf{X}, \mathbf{Y}) = \begin{cases} \sqrt{\frac{\mathcal{V}_n^2(\mathbf{X}, \mathbf{Y})}{\mathcal{V}_n^2(\mathbf{X}, \mathbf{X})\mathcal{V}_n^2(\mathbf{Y}, \mathbf{Y})}}, & \mathcal{V}_n^2(\mathbf{X}, \mathbf{X})\mathcal{V}_n^2(\mathbf{Y}, \mathbf{Y}) > 0, \\ 0, & \mathcal{V}_n^2(\mathbf{X}, \mathbf{X})\mathcal{V}_n^2(\mathbf{Y}, \mathbf{Y}) = 0. \end{cases}
$$

where $\mathcal{V}_n^2(\mathbf{X}, \mathbf{Y})$ is the empirical distance covariance with $\mathcal{V}_n^2(\mathbf{X}, \mathbf{Y}) = \frac{1}{n^2} \sum_{k,l=1}^n A_{kl} B_{kl}$. $A_{kl}$ and $B_{kl}$ are distance matrices defined by

$$
A_{kl} = a_{kl} - \bar{a}_k - \bar{a}_l + \bar{a}, \qquad \bar{a} = \frac{1}{n^2} \sum_{k,l=1}^n a_{kl}, \qquad \bar{a}_k = \frac{1}{n} \sum_{k=1}^n a_{kl}, \qquad \bar{a}_l = \frac{1}{n} \sum_{l=1}^n a_{kl}, \qquad a_{kl} = |X_k - X_l|_p
$$

with $|\circ|_p$ resembling the Euclidean norm in $\mathbb{R}^p$.

Distance correlation can be used to quantify the dependence between two sets of observations of differing dimensionality. In our case these two vectors are firstly a link strength or a underlying quantity of the networks (1d) and secondly the networks' position in the low dimensional embedding (2d). The resulting dependence value is used to rank the quantities in their ability to describe the structure of the low dimensional embedding.

## 2.6 Clustering and median network trajectories

On the reduced space we applied a clustering method named Ordering Points To Identify the Clustering Structure (OPTICS; Ankerst et al., 1999). OPTICS finds clusters by identifying regions of high density that contain a certain number of datapoints ($min_{samples}$). The cluster borders are defined by a certain drop in reachability of further datapoints ($max_{eps}$ and xi). This allows points that lie outside the reachability of neighbouring clusters to remain unclustered. The following settings were used for clustering: min_samples=80, max_eps=8 and xi=0.5. We calculated mean networks for each cluster by calculating the mean MCI value for each contemporaneous link among all networks contained in the cluster and only took those links that had an absolute value above 0.2.

## 2.7 Visualising ecosystem trajectories

As we calculated networks for each month for each measurement year for each FLUXNET site (if data requirements are fulfilled, see Sect. 2.3), annual trajectories can be visualised in the low dimensional embedding by connecting the dots representing the monthly networks of a specific year. Further, for each ecosystem, we calculated a monthly median trajectory within the t-SNE space which is composed of its monthly median networks. To this end, we calculated non-intersecting convex hulls which consisted of at least three datapoints (networks within the t-SNE space belonging to the same ecosystem, representing the same month, in at least three years). The monthly median network is the average of the networks lying on ($\geq 3$ networks) or in the inner hull ($< 3$ networks).

## 2.8 Workflow

Our restrictions on the data length and quality resulted in a selection of 119 FLUXNET sites (Fig. 1a). Applying above described procedure we obtained 10.038 networks for the different months and sites. An example network estimated by PCMCI is shown in Fig. 1c. The strongest and most consistent links are contemporaneous, indicating that interactions happen on time scales shorter than the time resolution. While lagged common drivers are excluded, contemporaneous links can still be spurious due to contemporaneous confounding (see Sect. 2.2). Nevertheless, we focus our analysis on these 15 links, as they contain most information. This is done by performing the dimensionality reduction on contemporaneous links and neglecting the lagged ones. The rational of employing a dimensionality reduction is the following. Each of the estimated networks constitutes one observation in a high dimensional space with a network's links spanning its axes (Fig. 1d). Projecting this high dimensional space onto two dimensions (Fig. 1e) allows first of all for visualisation. In case the data consists of a structure that can be 'identified' by the dimensionality reduction method, the visualisation reveals the dominant features of transitions between different states of biosphere–atmosphere interactions. The dominant features are the links that appear with strong gradients in the low dimensional embedding. To quantify and later rank the gradients exhibited by each link, we use the measure distance correlation (see Sect. 2.5). Therefore, we calculate the distance correlation of the link strengths (1d) with their position on the low dimensional embedding axes (2d). We also examine the distance correlation of secondary quantities with the axes. The secondary quantities are firstly mean values of variables calculated for each three month period of network estimation as well

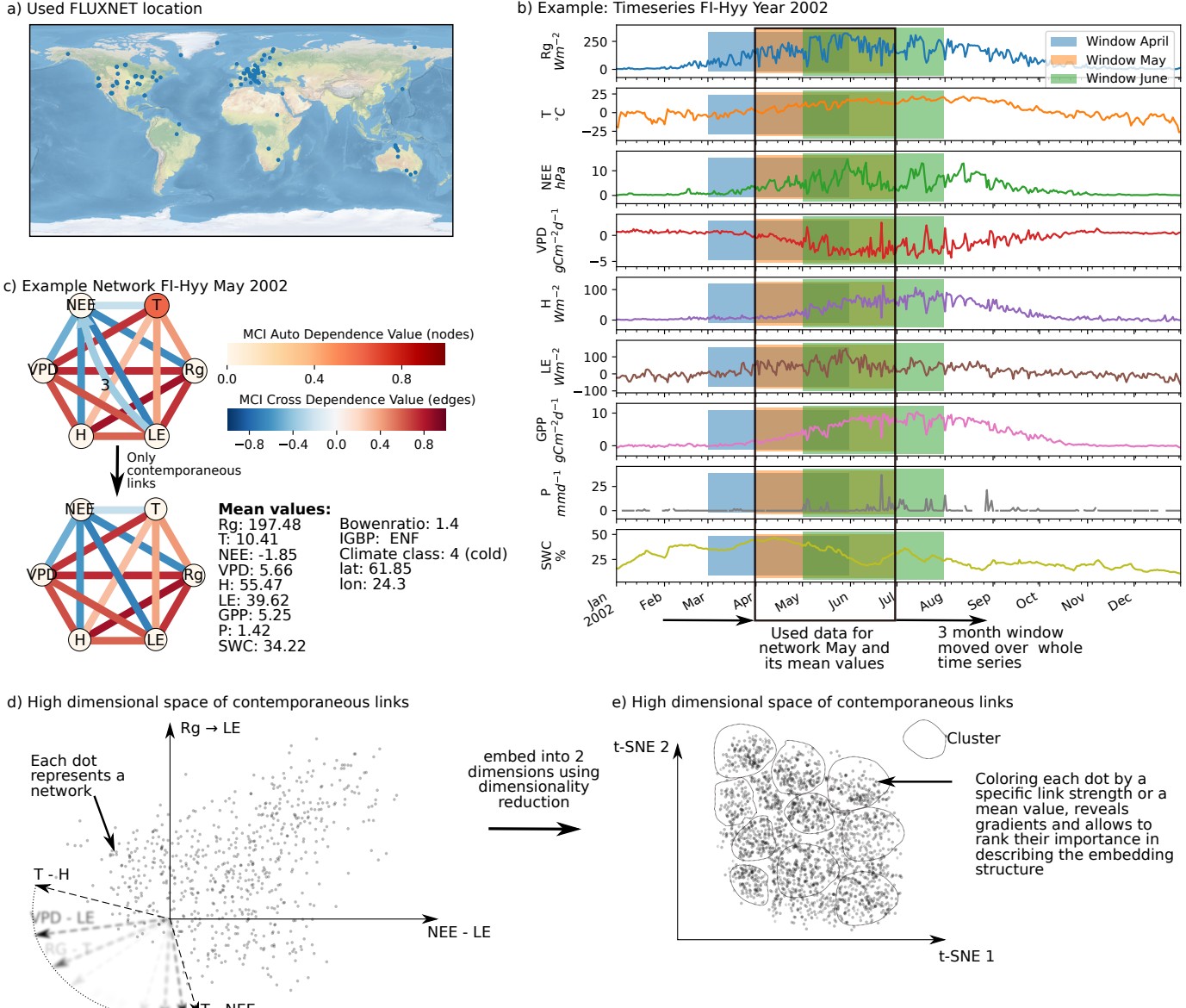

**Figure 1.** Schematic representation of the workflow. a) Eddy covariance data from the FLUXNET database are selected (119 sites). b) For each site we used the time series of global radiation R$_g$, air temperature T, vapour pressure deficit VPD, net ecosystem exchange NEE, sensible heat H, latent heat LE, gross primary productivity GPP, precipitation P and sail water content SWC. Networks are estimated in three month moving windows using Rg, T, NEE, VPD, LE and H. c) An example interaction network for FI-Hyy May 2002. Contemporaneous links are given by straight (undirected) edges, lagged link are given by curved arrows with a number indicating the time lag. The strongest and most persistent links are contemporaneous. Thus we limit our analysis to those links. d) Each three-month network can be interpreted as an observation in a 15-dimensional space (each contemporaneous link is one dimension). e) Dimensionality reduction projects all interaction networks into a two dimensional space preserving its local neighbourhood structure. Here any subsequent analysis and interpretation will be realised.

as secondly static values like climate class, vegetation type or location. The secondary quantities are used to find covariates of the low dimensional embedding that can help to explain its structure. In a next step, we cluster the low dimensional embedding to further understand to which network structures the gradients of link strength lead (see Sect. 2.4) and calculate the cluster's average networks (a simple mean). Up to this point (Sect. 3.1 and 3.2), we analysed the manifold of biosphere–atmosphere interactions and can address the first part of our hypothesis. As each point of the low dimensional embedding represents the biosphere–atmosphere interactions of a specific ecosystem at a specific time we can investigate the behaviour of specific ecosystems (see Sect. 2.7). Therefore we look at the monthly median and annual trajectories of certain ecosystems (Sect. 3.3 and 3.4). This leads to the answer of the second part of our hypothesis.

## 3    Results and Discussions

### 3.1    Two-dimensional embedding of biosphere–atmosphere networks

To find the most suitable dimensionality reduction method, we evaluated three different methods (PCA, t-SNE and UMAP) with respect to their ability to project the high dimensional network space onto two dimensions. To compare the low-dimensional embedding spaces, we used the $R_{NX}(K)$ measure (see Sect. 2.4) which quantifies how well neighbourhoods are preserved when projecting the high dimensional space onto fewer dimensions. We found that t-SNE achieved the best projection, by best preserving both local and distant neighbourhoods (cf. Sect. 2.4, Fig. A1, B1). This is unexpected as UMAP is said to intentionally preserve the global structure. Yet, as can be seen in Fig. 4a, the networks almost form a continuum. Thus, by maintaining the local neighbourhood structure, also the global structure is preserved within t-SNE.

The two-dimensional embedding by t-SNE of biosphere–atmosphere interactions is ordered primarily by dependencies including carbon flux (NEE) and energy distributions (LE, H). This can be seen in Fig. 2 which shows the 2d embedding colour-coded by the strength of individual links, i.e. MCI partial correlation values. The colouring reveals that the link strengths are ordered along gradients, i.e. they exhibit some dependence with the t-SNE axes. Using distance correlation to rank those gradients (see Sect. 2.5), we find the links NEE–LE ($\mathcal{R} = 0.75$), Rg–LE ($\mathcal{R} = 0.73$) and T–H ($\mathcal{R} = 0.69$) to have the strongest gradients. The connection between carbon and water fluxes as well as the role of energy input to sustain water fluxes (if available in the soil) are well known and investigated dependencies (Beer et al., 2010; Luyssaert et al., 2007).

To search for covariates that help to explain - and if thought further, help to predict the network structures- we colour coded the embedding by the networks' underlying mean conditions, i.e. the average over the respective time window, of the exchange rates (GPP, NEE, LE and H) as well as meteorological conditions (Rg, T, VPD, P). This is shown in Fig. 3. Clearly, the mean exchange rates and meteorological conditions - although not considered in the estimation of the networks - are related to the observed biosphere–atmosphere interactions. On the contrary, corresponding vegetation types and Köppen-Geiger classes are not as much related as displayed in the Supplementary Material section Fig. E2. The results show that a high dimensional space encompassing more than 10000 ecosystem networks representing the states of biosphere–atmosphere interactions from ecosystems of various geographic origins can be reduced to a compact two dimensional manifold characterised by four edges and gradients of mean biosphere and atmosphere conditions. While gradients in MCI partial correlation strength are expected

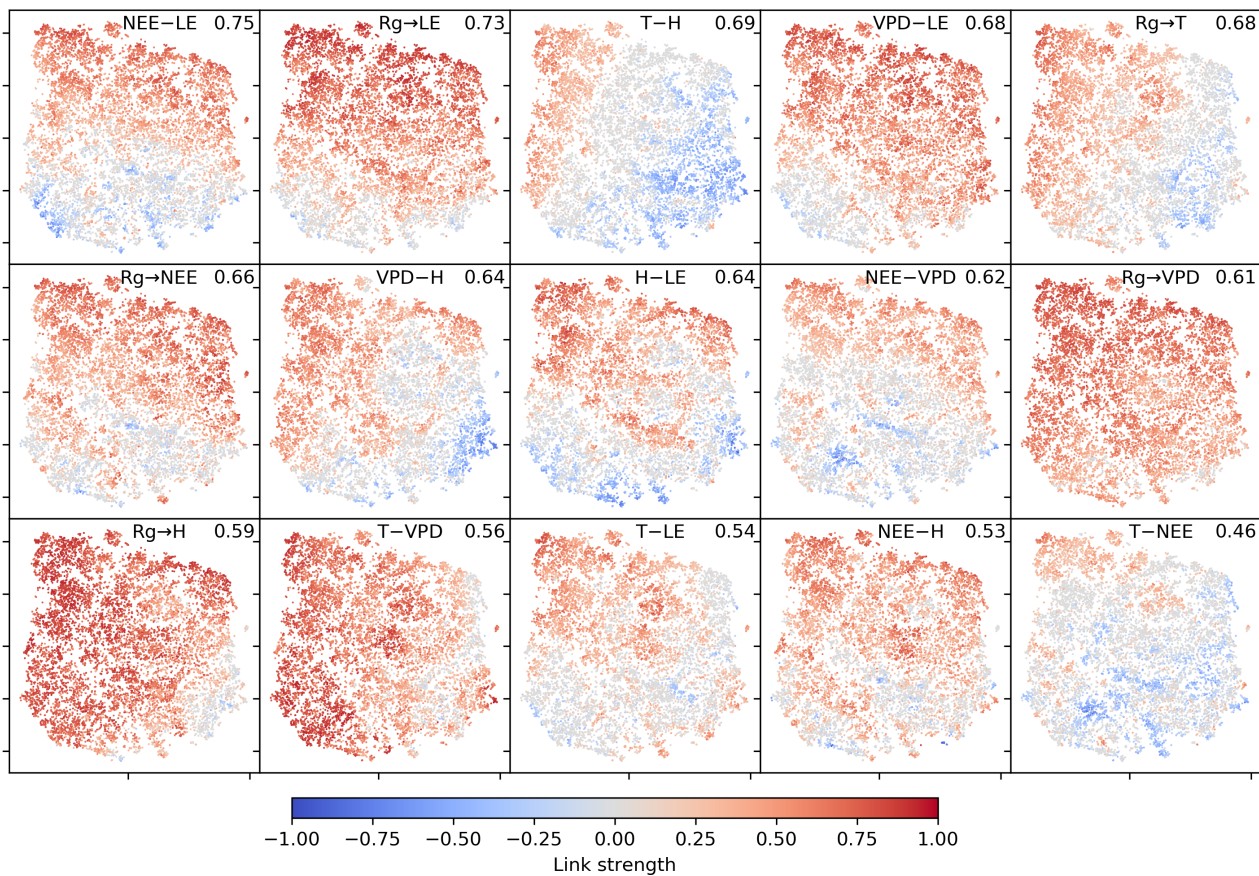

**Figure 2.** Two-dimensional embedding of three-monthly biosphere–atmosphere networks realised via t-SNE. Shown is the distribution of link strengths among the networks. The strength is estimated via MCI partial correlation values. Subfigures are sorted by the distance correlation of the link's MCI value with the axes (value in upper right corner). As $R_g$ is set as potential driver (PCMCI parameter 'selected_links', see table B1), connections including $R_g$ are directed $\rightarrow$. This setting does not affect the results (see Fig. D1).

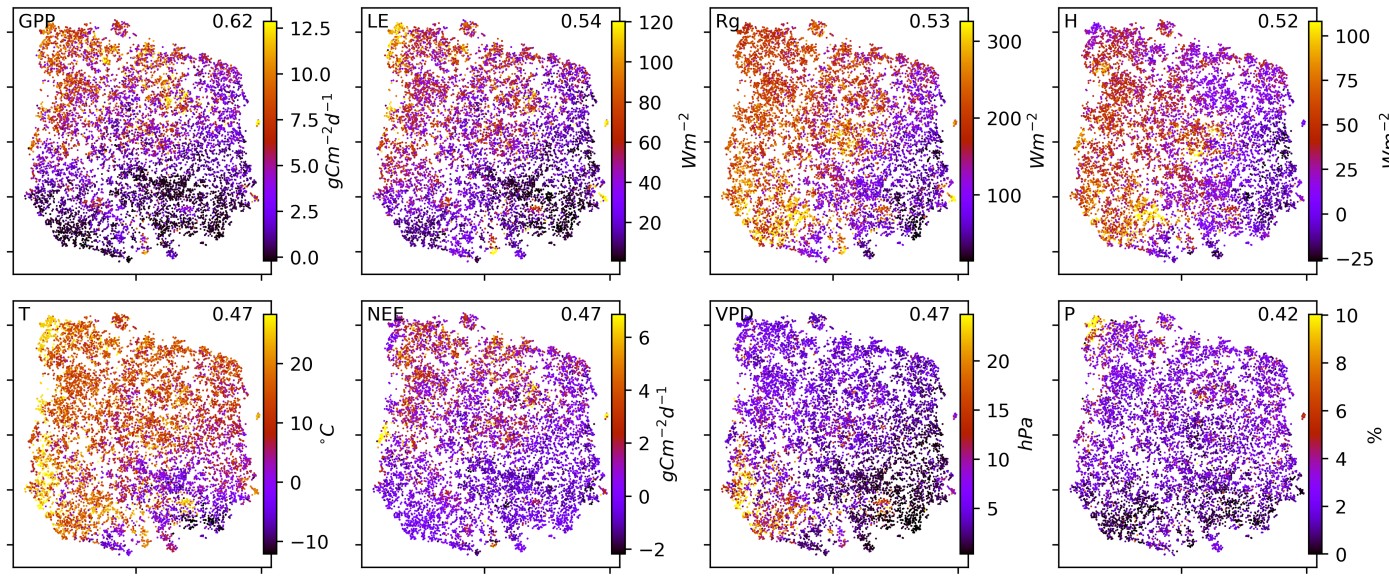

**Figure 3.** Two-dimensional embedding coloured by underlying mean exchange rates and meteorological conditions. The mean values are calculated over the respective time periods used for the network estimation. Each network is estimated on a three month window of daily time series data. Values are cut off at the highest and lowest percentile. Distance correlation of the shown quantity with the axes is given in upper right corner of each panel.

as they were used as features in the dimensionality reduction, gradients in mean climatic and biospheric conditions were not. This information thus must be entailed in the networks' structure. To better grasp the distribution of network structures, we further analyse the emerging clusters.

### 3.2 Clusters of characteristic ecosystem–atmosphere networks

As we apply a significance threshold to each link of the estimated network structures (see Sect. 2.3), the networks typically lack weak links. This leads to a certain degree of clustering among the networks, which we identified using the OPTICS approach (see Sect. 2.6; Ankerst et al., 1999)) (Fig. 4a). Cluster boundaries are shown by the convex hulls in Fig. 4b, where we also visualise the mean networks of each cluster. This visualisation reveals that the mean networks of the clusters situated at the embedding's edges can be regarded as archetypes of network structures, i.e. extremal, characteristic states (similar to the concept of endmember states). The four states can be described as follows:

**Type 1** is a sparsely connected network. Links, if present, are very weak and predominantly exist among atmospheric variables. Mean atmospheric conditions are characterised by low energy input (low $R_g$ and T). Carbon and water fluxes are consequently close to zero, and daily averages of sensible heat can even reach negative values. Such conditions reflect high latitude ecosystem winter states experienced by ecosystems like the evergreen needle leaf forests (ENF) of Finnland, i.e. Hyytiälä (FI-Hyy) and Sodankyla (FI-Sod) as well as Canada, i.e., the UCI-1850 burn site (CA-NS1) and Quebec - Eastern Boreal (CA-Qcu) during December and January.

**Type 2** consists of strong links among atmospheric variables but LE and NEE are weakly, not, or even negatively connected to the atmosphere, i.e. the meteorological variables. This network structure coincides with high energy input (high $R_g$ and T) but low water availability (low P and SWC, high VPD). A high Bowen ratio, i.e. the ratio between sensible heat and latent heat, representing aridity, and low absolute carbon fluxes (GPP and NEE) are the consequence. These conditions are typically present at semi-arid ecosystems like the woody savanna (WSA) Santa Rita Mesquite (US-SRM) as well as the grasslands Santa Rita (US-SRG), Audubon Research Ranch (US-Aud) and Sturt Plains (AU-Stp) during dry season.

**Type 3** exhibits the same strong links among Rg, VPD and H as Type 2 but T is weakly or not connected. The opposite is true for links of LE and NEE which are strongly connected to the other variables (except T). $R_g$ and T are considerably lower than in Type 2 (approximately by 100 W/m$^2$ and 10°C) but because of sufficient water availability the Bowen ratio is between 0 and 1. Typical ecosystems in this state are mid to high latitude forests during spring or autumn, e.g. Harvard Forest EMS Tower (US-Ha1, deciduous broadleaf forest (DBF)), Roccarespampani 1 (IT-Ro1, DBF), Vielsalm (BE-Vie, mixed forest (MF)) and Hyytiälä (FI-Hyy, ENF).

**Type 4** is fully and strongly connected. Both energy input and water availability are high leading to Bowen ratios around 1. This network state is typically present in tropical forests like the Guyaflux site in French Guiana (GF-Guy) (evergreen broadleaf forest (EBF)) but can temporarily be also reached by a variety of other ecosystems, e.g. mid and high latitude forests like Hainich (DE-Hai, DBF), Tharandt (DE-Tha, ENF), BE-Vie (MF), FI-Hyy (ENF) as well as woody savannas (WSA) as Howard Springs (AU-How) and grasslands as Daly River Savanna (AU-Dap).

The archetypes of networks are located at the edges of the two-dimensional space and thus could define two imaginary axes. From a physical point of view, energy is required for each process and interaction to occur, e.g. photosynthesis or evaporation (Bonan, 2015). Therefore, transitions along the axis connecting the network types 1 and 4 might be interpreted as energy controlled as dependencies among all variables fade or increase consistently. Transitions along the axis connecting network types 2 and 3 are explainable by a combination of water availability and a temperature gradient. Low water availability but high temperatures cause shut down of stomatal conductance or ecosystems to enter a dormant state which leads to low carbon and water fluxes and low connectivity. On the other hand, sufficient water and medium temperatures (around the optimum of photosynthesis) allow for carbon and water fluxes but reduce the influence of varying temperatures leading to connected NEE and LE but disconnected T. And indeed these patterns and gradients exist. Mean $R_g$ is lowest at network type 1 and almost linearly increases towards network type 4. P is lowest at network type 1 and 2. In combination with high energy input network

type 2 has lowest SWC values and the highest Bowen ratios (see Supplementary Material section Fig. E2). SWC is higher but quite dispersed elsewhere suggesting that at a certain point water limitations are fading out. T values of course also show an increase from network type 1 to 4 (as radiation) but also from network type 3 to 2 and are actually rather low (8°C to 15 °C) at network type 3 (see Fig 3). As meteorological conditions affect biosphere productivity, network type 1 and 2 exhibit low, type 3 medium and type 4 high productivity i.e. estimated as GPP. In short, the clustering revealed that changes in energy and water availability can explain major transitions between different states of biosphere–atmosphere interactions. This is in line with a recent study showing that a variety of land-surface processes can be largely summarised by on the one hand productivity measures and on the other hand water and energy availability. Both, water and energy availability, need to be high for high productive states, yet the lack of either of them leads to low productivity (Kraemer et al., 2020a). This biosphere state triangle is found in our analysis by the network type 1 (cold - low connectivity), 2 (dry - NEE/LE weakly connected) and 4 (high productivity - fully connected). Yet, a fourth network type (type 3) is naturally occurring in the t-SNE space as we here include interactions with the atmosphere.

Up to this point we have found strong evidence supporting our first hypothesis. The manifold of biosphere–atmosphere interactions can be represented rather well by two dimensions which we identified to be most consistent with energy and water availabilities. It is confined by four characteristic states and populated homogeneously by the observed network states. Having an understanding of the low dimensional embedding's structure now allows us to analyse specific ecosystem behaviour.

### 3.3 Ecosystems' median trajectories

Each point in the reduced t-SNE space represents a biosphere–atmosphere interaction network for a given month and ecosystem. Hence, we can trace an ecosystem's trajectory through time. We are first focusing on an ecosystem's median monthly trajectory (see Sect. 2.7) within the low dimensional space. We can see that the median trajectories reflect seasonal patterns of meteorological conditions (Fig. 5). For example, mid-latitude sites like FR-Pue (EBF), DE-Hai (DBF) and FI-Hyy (ENF) exhibit a strong seasonal variation of $R_g$ and span a long distance in the t-SNE space. In contrast, tropical ecosystems like GF-Guy (EBF) constantly have high $R_g$ and exhibit predominantly network type 4 indicative of high productive conditions - while DE-Hai or FI-Hyy reach this connectivity pattern only during peak growing season. US-SRM (WSA), however, has similar or even higher $R_g$ values throughout the year but barely manages to deviate from type 2 which is in accordance with its low water availability. The amount of precipitation further aligns with differences and characteristics of the trajectories of FR-Pue, DE-Hai and FI-Hyy. For example, FI-Hyy shows some deviation towards edge 2 in February and March, FR-Pue in June, July and August. For both, mean precipitation is lowest during these months. These behaviours demonstrate what the previous figures (Fig. 3 and 4) have already suggested: Ecosystem's populate the low dimensional space and migrate within as allowed by their climatic conditions. Thereby they can exhibit a wide range of interaction structures as can be seen from the mid-latitude sites. As these behaviours are multi year averages they could resemble more ecosystem adaptation to median climatic conditions than flexible adjustment of biosphere–atmosphere interactions to quickly changing meteorological conditions. If biosphere–atmosphere interactions are confined by adaptation shall be investigated in the final analysis section.

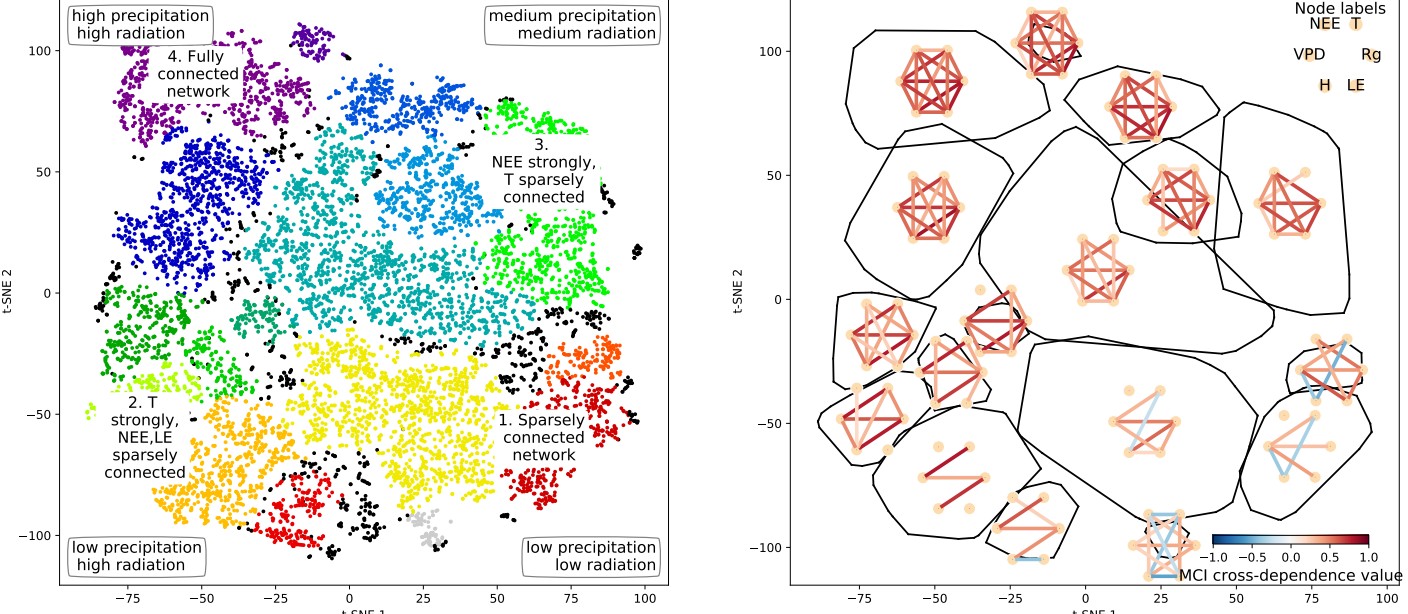

**Figure 4.** Structure of the two-dimensional embedding. left: t-SNE space clustered by the OPTICS approach (Ankerst et al., 1999). Colours represent different clusters, black dots are not attributed to a cluster. Indicated are the four archetypes of network connectivity and the networks underlying meteorological conditions. right: Convex hulls of clusters and their average network, i.e. average over all networks belonging to one cluster. Average networks are thresholded at a minimum link strength of 0.2. A finer clustering can be found in the Supplementary Material section Fig. E1.

### 3.4 Deviations from ecosystem median trajectories

The remaining open question is, how flexible do the networks adjust to deviations from mean climatic conditions. Therefore, we look at climatic anomalies. Figure 6 shows the trajectories of ecosystems during anomalous dry or wet conditions. During the European heatwave of 2003, in July and August the trajectories of two temperate central European forests, DE-Hai and DE-Tha, no longer manage to establish a network structure resembling network type 4, typical for these ecosystems during their high productive phase. Instead they are shifted towards network type 2, associated with drier conditions (Fig. 6a, b). Similarly, the ecosystem BR-Sa3 (EBF) in the Brazilian tropical rainforest shows substantial deviations towards network type 2 during the exceptional dry season of 2001 (Aug, Sep, Oct) (Marengo et al., 2018) (Fig. 6c). In contrast, US-Wkg is a grassland accustomed to dry conditions and thus predominantly exhibits low water and carbon fluxes resulting in network structures as of network type 2, i.e. water and carbon fluxes are barely or even disconnected. Carbon and water fluxes of semi-arid ecosystems, however, are known to respond quickly and strongly to sufficient precipitation (Potts et al., 2019; Leon et al., 2014; Reynolds et al., 2004). This sensitivity is found to carry over to the network structure as well. The network structure of US-Wkg becomes fully connected (network type 4) in September 2014 with above average precipitation (NOAA) (Fig. 6d). Summarising, climatic

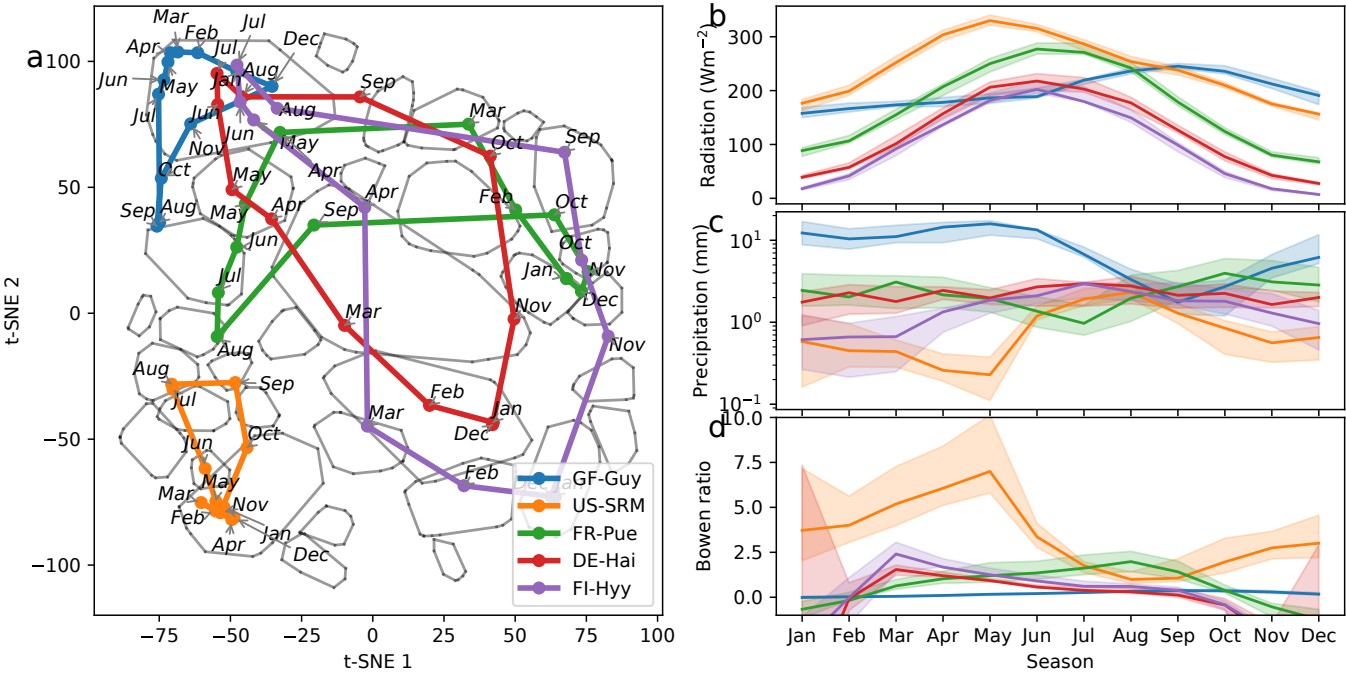

**Figure 5.** Median trajectories of selected sites (left) and their corresponding mean values of radiation, precipitation and the Bowen ratio (right). In winter month the Bowen ratio can turn negative. Nevertheless we set the lower limit of the y-axis to 0. As networks are calculated using a centred three month moving window, each month is ascribed a network. Thus, the behaviour of an ecosystem can be tracked by its monthly networks, which form trajectories for each year. An ecosystem's monthly median trajectory is composed of the two dimensional monthly median networks (see Sect. 2.7 for details).

extremes are visible in an ecosystem's trajectory as strong deviations from the median trajectory. With this finding we have to reject our second hypothesis that owing to an ecosystem's adaptation its accessible functional states are limited to a certain range. The opposite seems to be valid. Biosphere–atmosphere interactions can follow flexibly atmospheric conditions and are not confined to certain states.

### 3.5 Functional convergence of biosphere–atmosphere interactions

We have seen that networks representing biosphere–atmosphere interactions strongly align with prevailing mean meteorological conditions. Moreover, the visualisation of ecosystem trajectories within the t-SNE space (Fig. 5, 6) and the distributions of vegetation types and climatic regions (Supplementary Material Fig. E2) reveal that ecosystems across vegetation types and climatic regions can exhibit similar biosphere–atmosphere interactions if their meteorological conditions are similar. For example, we found a fully connected network (type 4) to be associated with high radiation and water availability and thus optimal growing conditions, which results in high carbon and water fluxes. Diverging from optimal growing conditions, links in the

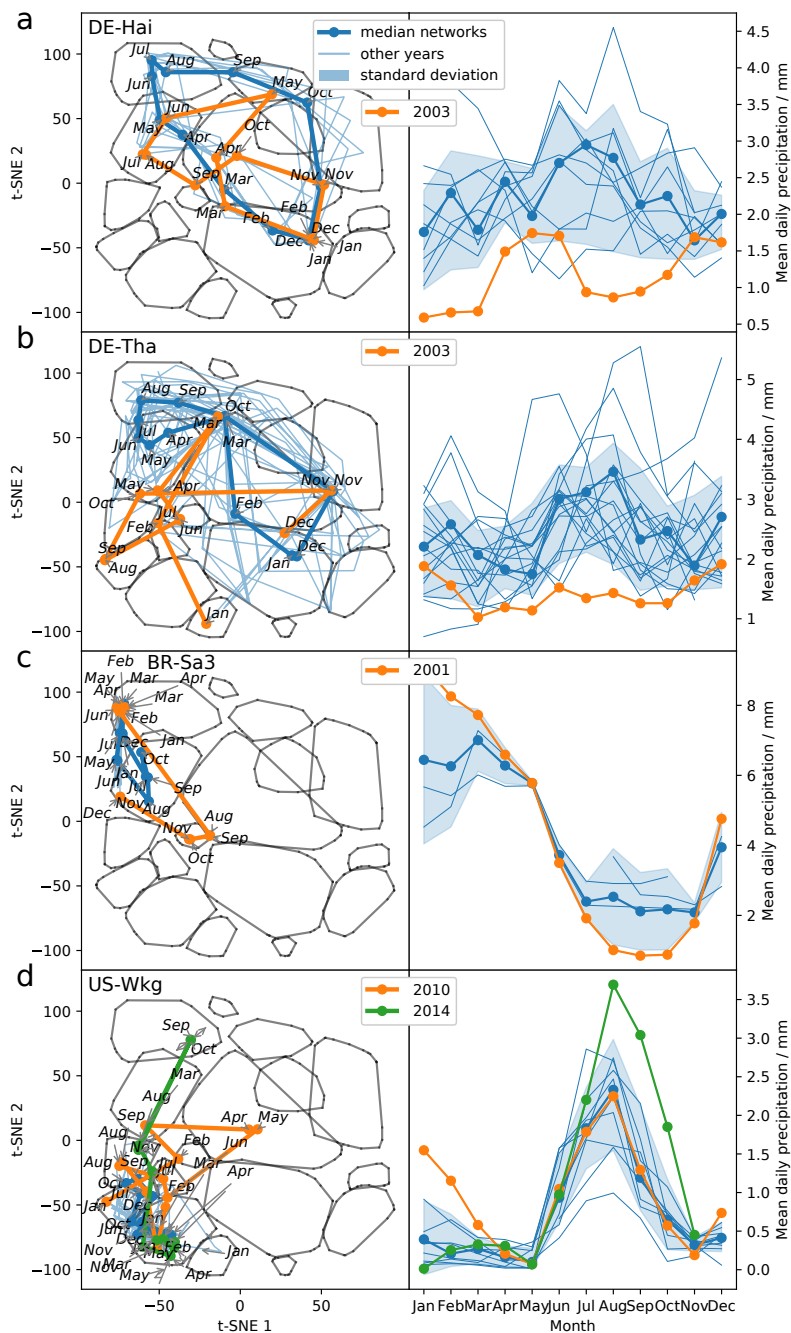

**Figure 6.** Abnormal conditions in meteorological conditions (here precipitation) become visible in an ecosystem's trajectory. left: Trajectories within the low dimensional space of the ecosystems Hainich (DE-Hai, DBF), Tharand (DE-Tha, ENF), Santarem-Km83-Logged Forest (BR-Sa3, EBF) and Walnut Gulch Kendall Grasslands (US-Wkg, GRA). right: Three monthly average of daily precipitation data.

networks weaken and disappear. This behaviour can be understood as functional convergence of ecosystems which corroborates the hypothesis that ecosystems have a low number of key processes that determine ecosystem behaviour (Lambert, 2006; Meinzer, 2003; Shaver et al., 2007) rendering their behaviour transparent and predictable. Criticism might rise as the larger part of the biosphere–atmosphere interaction network indeed is a pure atmospheric network, i.e. Rg, T, VPD and H. Thus strong associations of networks and their trajectories with atmospheric conditions could be dominated by changes in this atmospheric network. Fig. 2, however, suggests the opposite. The strongest gradients are given by the links NEE–LE and Rg–LE and transitions along the axis connecting type 2 and 3 (cf. Fig. 4) are dominated by changes in biosphere connectivity, i.e. LE and NEE.

In fact, the dominance of climatic drivers in controlling the temporal evolution of ecosystem functioning emerges also in other studies (Musavi et al., 2017; Schwalm et al., 2017; Kraemer et al., 2020a) as they showed that carbon fluxes are primarily controlled by climatic factors. Yet, these and others also show the role of biotic factors in shaping the responses of ecosystem processes to climatic variability. For example, Musavi et al. (2017) revealed in a global ecosystem study that species diversity and ecosystem age decrease inter annual variability of GPP. Similarly, Wagg et al. (2017) showed biodiversity to increase long-term stability of ecosystem productivity. In regional studies Wales et al. (2020) found the stability of net primary production to be affected by the kind and severity of disturbances. Tamrakar et al. (2018) showed that seasonal carbon fluxes were more sensitive to environmental conditions in a homogeneous forest compared to a heterogeneous one. It would be of interest to investigate, to which degree the effects of biotic factors also translates to the sensitivity of the network structure.

Furthermore, extreme heat and drought events (Sippel et al., 2018) or compound events in general (Zscheischler et al., 2020) can severely disrupt ecosystem functions. The time of recovery from such disturbances is a crucial parameter in assessing ecosystem resilience. Schwalm et al. (2017) showed that the recovery time measured as the recovery in GPP is primarily influenced by climate but secondarily by biodiversity and $CO_2$ fertilisation. Assessing the recovery time via GPP already puts the ecosystem functioning into focus. The here presented framework, i.e. the sensitivity of an ecosystem's network structure to meteorological conditions, might be a valuable asset to study reaction time and strength to and recovery from extreme events as it not only utilises one variable but the interactions of a set of variables, thereby capturing more comprehensively an ecosystem state. A drawback is the reduced temporal resolution (a certain time period of daily or even half hourly measurements is aggregated to one network) which can be offset by the here used moving window approach to a certain degree. Especially with regard to climatic extreme conditions in recent years with observed vegetation dieback in, for example, DE-Hai (Schuldt et al., 2020), further studies could also shed light on the role of adaptation in shaping biosphere–atmosphere interactions. Our study suggest that adaptation to a lesser degree limits the range of possible interactions but enables to sustain and persist certain conditions for longer periods. The focus of further studies thus could be to elucidate the role of biotic factors in influencing ecosystem trajectories as well as the role of adaptation and the response to extreme events.

## 3.6 Limitations of the study

Finally, we would like to take a critical view on our analysis approach. As stated in Sect. 2.2, PCMCI might fail to identify some spurious links due to the occurrence of contemporaneous confounders. Thus networks can not be interpreted causally

but this does not severely hinder their value for the current analysis. In addition we include a rather limited set of variables into the network estimation. Thus we cannot and do not claim that ecosystems become fully alike under similar meteorological conditions. Yet, on the timescale investigated the data shows, that the interactions among the chosen set of variables can be described by very similar structures. Follow up studies might search for and include further biosphere variables. Currently, an analysis of the biotic effects on the network structure is hampered because the t-SNE space is not metric. Thus, for instance, the effect of a drought with similar magnitude in a boreal and temperate forest cannot simply be compared by the deviation from their median trajectory.

## 4   Conclusions

We analysed the functional behaviour of a variety of ecosystems using the FLUXNET database of carbon, water, and energy flux measurements. In particular, we examined the interaction structure between biosphere–atmosphere fluxes as well as atmospheric state variables using PCMCI, a method to estimate causal relationships from empirical time series under certain assumptions. Using non-linear dimensionality reduction, we find evidence supporting our hypothesis that the manifold of existing states is bound by few, i.e. four, archetypes of network states. They are characterised on the one hand by a fully connected and almost unconnected network structure and on the other hand by an antagonistic coupling of carbon and water flux with temperature - when one is strongly coupled, the other is decoupled. The transitions between these states correlate well with gradients of meteorological drivers, i.e. radiation and water availability. The movement of an ecosystem within that space therefore strongly aligns with changes in meteorological conditions. This, however, also leads to similar behaviour under similar conditions for strongly contrasting ecosystems. For example, forests of mid or even high latitudes exhibit similar interaction structure as tropical forests given high radiation and water availability during summer. Yet, this state can also be reached by predominantly dry ecosystems like steppe grasslands given sufficient precipitation. In contrast if productive ecosystems are struck by a severe drought, like central European ecosystems in 2003, the behaviour can adapt more to that of a Mediterranean ecosystem. Thus the second part of our hypothesis must be rejected. The analysis shows that the biosphere-atmosphere interaction structure can adapt flexibly to prevailing conditions and is widely independent of vegetation type and climatic region. Such behaviour is strong evidence for functional convergence of ecosystems, i.e. their behaviour is determined by a low number of key processes. For further studies, we suggest, to focus on the role of biotic factors as, for example, plant functional types, ecosystem age and adaptation. These factors could play crucial roles in understanding the ecosystem copying strategies to climatic extremes.

*Code availability.*   Code scripts can be found at https://github.com/ckrich/Functional-convergence-of-biosphere-atmosphere-interactions-in-response-to-meteorology

*Data availability.* The eddy covariance data of the FLUXNET sites can be downloaded from the official webpage (https://fluxnet.fluxdata.org/).

## Appendix A: Methods

Table A1: List of FLUXNET sites used for the generation of artificial datasets and the time period used.

| FLUXNET-ID | IBGP | Koeppen-Geiger Class | start year | end year | data reference |
|---|---|---|---|---|---|
| AT-Neu | GRA | Dfb | 2002 | 2012 | Wohlfahrt et al. (2008) |
| AU-ASM | ENF | BSh | 2010 | 2014 | Cleverly et al. (2013) |
| AU-Cpr | SAV | Csa | 2010 | 2014 | Meyer et al. (2015) |
| AU-DaP | GRA | Aw | 2007 | 2013 | Beringer et al. (2011a) |
| AU-DaS | SAV | Aw | 2008 | 2014 | Hutley et al. (2011) |
| AU-Dry | SAV | | 2008 | 2014 | Cernusak et al. (2011) |
| AU-How | WSA | Aw | 2001 | 2014 | Beringer et al. (2007) |
| AU-Stp | GRA | Aw | 2008 | 2014 | Beringer et al. (2011b) |
| AU-Tum | EBF | Cfb | 2001 | 2014 | Leuning et al. (2005) |
| AU-Wom | EBF | Cfb | 2010 | 2014 | Arndt et al. |
| BE-Bra | MF | Cfb | 1996 | 2014 | Carrara et al. (2004) |
| BE-Lon | CRO | BSk | 2004 | 2014 | Moureaux et al. (2006) |
| BE-Vie | MF | Cfb | 1996 | 2014 | Aubinet et al. (2001) |
| BR-Sa3 | EBF | | 2000 | 2004 | Saleska et al. (2003) |
| CA-Mer | WET | Dwb | 1998 | 2005 | Lafleur et al. (2003) |
| CA-NS1 | ENF | BWk | 2001 | 2005 | Goulden et al. (2006) |
| CA-NS2 | ENF | BWk | 2001 | 2005 | Bond-Lamberty et al. (2004) |
| CA-NS3 | ENF | | 2001 | 2005 | Wang et al. (2002a) |
| CA-NS5 | ENF | BSk | 2001 | 2005 | Wang et al. (2002b) |
| CA-NS6 | OSH | BSk | 2001 | 2005 | Wang et al. (2002c) |
| CA-Qcu | ENF | Dwb | 2001 | 2006 | Giasson et al. (2006) |
| CA-Qfo | ENF | Dfb | 2003 | 2010 | Chen et al. (2006) |
| CA-SF2 | ENF | BSk | 2001 | 2005 | Rayment and Jarvis (1999a) |
| CA-SF3 | OSH | Dwc | 2001 | 2006 | Rayment and Jarvis (1999b) |
| CH-Cha | GRA | Cfb | 2005 | 2014 | Merbold et al. (2014) |
| CH-Dav | ENF | Dfc | 1997 | 2014 | Zielis et al. (2014) |
| CH-Fru | GRA | Dfb | 2005 | 2014 | Imer et al. (2013) |
| CH-Lae | MF | BWk | 2004 | 2014 | Etzold et al. (2011) |

*Continued on next page*

| FLUXNET-ID | IBGP | Koeppen-Geiger Class | start year | end year | data reference |
|---|---|---|---|---|---|
| CH-Oe1 | GRA | Cfb | 2002 | 2008 | Ammann et al. (2009) |
| CH-Oe2 | CRO | BSk | 2004 | 2014 | Dietiker et al. (2010) |
| CZ-BK1 | ENF | Dwb | 2004 | 2014 | Acosta et al. (2013) |
| CZ-BK2 | GRA | Dfb | 2004 | 2012 | Sigut et al. |
| CZ-wet | WET | Dfb | 2006 | 2014 | Dušek et al. (2012) |
| DE-Akm | WET | BWk | 2009 | 2014 | Bernhofer et al. (a) |
| DE-Geb | CRO | Cfb | 2001 | 2014 | Anthoni et al. (2004b) |
| DE-Gri | GRA | Dfb | 2004 | 2014 | Prescher et al. (2010a) |
| DE-Hai | DBF | Cfb | 2000 | 2012 | Knohl et al. (2003) |
| DE-Kli | CRO | Dfb | 2004 | 2014 | Prescher et al. (2010b) |
| DE-Lkb | ENF | Dwb | 2009 | 2013 | Lindauer et al. (2014) |
| DE-Obe | ENF | Dfb | 2008 | 2014 | Bernhofer et al. (b) |
| DE-Spw | WET | BWk | 2010 | 2014 | Bernhofer et al. (c) |
| DE-Tha | ENF | Dfb | 1996 | 2014 | Grünwald and Bernhofer (2007) |
| DE-Wet | ENF | Dfb | 2002 | 2006 | Anthoni et al. (2004a) |
| DK-NuF | WET | Dfc | 2008 | 2014 | Westergaard-Nielsen et al. (2013) |
| DK-Sor | DBF | Cfb | 1996 | 2014 | Pilegaard et al. (2011) |
| DK-ZaH | GRA | ET | 2000 | 2014 | Lund et al. (2012) |
| ES-ES1 | ENF | BSk | 1999 | 2006 | Sanz et al. (2004) |
| FI-Hyy | ENF | Dfb | 1996 | 2014 | Suni et al. (2003) |
| FI-Kaa | WET | Dfc | 2000 | 2006 | Aurela et al. (2007) |
| FI-Sod | ENF | BSk | 2001 | 2014 | Thum et al. (2007) |
| FR-Fon | DBF | Cfb | 2005 | 2014 | Delpierre et al. (2016) |
| FR-Gri | CRO | Cfb | 2004 | 2014 | Loubet et al. (2011) |
| FR-Hes | DBF | Cfb | 1997 | 2006 | Granier et al. (2000) |
| FR-LBr | ENF | Cfb | 1996 | 2008 | Berbigier et al. (2001) |
| FR-Pue | EBF | Csa | 2000 | 2014 | Rambal et al. (2004) |
| GF-Guy | EBF | Am | 2004 | 2014 | Bonal et al. (2008) |
| HU-Bug | GRA | Dfb | 2002 | 2006 | Nagy et al. (2005) |
| IL-Yat | ENF | BWh | 2001 | 2006 | Grünzweig et al. (2003) |
| IT-Amp | GRA | Dsb | 2002 | 2006 | Gilmanov et al. (2007) |
| IT-BCi | CRO | Csa | 2004 | 2014 | Vitale et al. (2016) |

| FLUXNET-ID | IBGP | Koeppen-Geiger Class | start year | end year | data reference |
|---|---|---|---|---|---|
| IT-Col | DBF | Dsb | 1996 | 2014 | Valentini et al. (1996) |
| IT-Cpz | EBF | Csa | 1997 | 2009 | Garbulsky et al. (2008) |
| IT-Lav | ENF | Dwb | 2003 | 2014 | Marcolla et al. (2003) |
| IT-MBo | GRA | Dfb | 2003 | 2013 | Marcolla et al. (2011) |
| IT-Noe | CSH | BSk | 2004 | 2014 | Reichstein et al. (2002) |
| IT-Non | DBF | Cfa | 2001 | 2006 | Nardino et al. (2002) |
| IT-Ren | ENF | BSk | 1998 | 2013 | Marcolla et al. (2005) |
| IT-Ro1 | DBF | Csa | 2000 | 2008 | Rey et al. (2002) |
| IT-Ro2 | DBF | Csa | 2002 | 2012 | Tedeschi et al. (2006) |
| IT-SRo | ENF | BSk | 1999 | 2012 | Chiesi et al. (2005) |
| IT-Tor | GRA | BSk | 2008 | 2014 | Galvagno et al. (2013) |
| JP-SMF | MF | Cfa | 2002 | 2006 | Matsumoto et al. (2008) |
| NL-Hor | GRA | Csb | 2004 | 2011 | Jacobs et al. (2007) |
| NL-Loo | ENF | Cfb | 1996 | 2014 | Moors (2012) |
| PT-Esp | EBF | Csa | 2002 | 2006 | Rodrigues et al. (2011) |
| RU-Cok | OSH | Dwd | 2003 | 2014 | van der Molen et al. (2007) |
| RU-Fyo | ENF | Dwb | 1998 | 2014 | Kurbatova et al. (2008) |
| SD-Dem | SAV | BWh | 2005 | 2009 | Ardö et al. (2008) |
| SE-Deg | WET | Dwc | 2001 | 2005 | Sagerfors et al. (2008) |
| SE-Fla | ENF | Dwc | 1996 | 2002 | Valentini et al. (2000) |
| SE-Nor | EBF | BSk | 1996 | 2005 | Lagergren et al. (2008) |
| UK-Gri | ENF | Csb | 1997 | 2006 | Medlyn et al. (2005) |
| US-ARM | CRO | Csa | 2003 | 2012 | Fischer et al. (2007) |
| US-Aud | GRA | BSk | 2002 | 2006 | - |
| US-Blo | ENF | Csb | 1997 | 2007 | Schade et al. |
| US-Bo1 | CRO | Dfa | 1996 | 2007 | Meyers and Hollinger (2004) |
| US-Cop | GRA | BWk | 2001 | 2007 | Ruehr et al. (2012a) |
| US-FPe | GRA | BSk | 2000 | 2006 | Gilmanov et al. (2005) |
| US-GBT | ENF | BWk | 1999 | 2006 | Zeller and Hehn (1996) |
| US-GLE | ENF | Dsc | 2004 | 2014 | Zeller and Nikolov (2000) |
| US-Ha1 | DBF | Dfb | 1991 | 2012 | Wofsy et al. (1993) |
| US-Ho1 | ENF | Dfb | 1996 | 2004 | Armstrong and Ernst (1999) |

| FLUXNET-ID | IBGP | Koeppen-Geiger Class | start year | end year | data reference |
|---|---|---|---|---|---|
| US-Los | WET | Dfb | 2000 | 2014 | Baker et al. (2003) |
| US-MMS | DBF | Dfa | 1999 | 2014 | Pryor et al. (1999) |
| US-Me2 | ENF | Dsb | 2002 | 2014 | McDowell et al. (2004) |
| US-Me6 | ENF | BSk | 2010 | 2014 | Ruehr et al. (2012b) |
| US-Myb | WET | Csb | 2010 | 2014 | Ruehr et al. (2012c) |
| US-NR1 | ENF | Dfc | 1998 | 2014 | Reich et al. (1998) |
| US-Ne1 | CRO | Dwa | 2001 | 2013 | Gitelson et al. (2003) |
| US-Ne2 | CRO | Dwa | 2001 | 2013 | Cassman et al. (2003a) |
| US-Ne3 | CRO | Dwa | 2001 | 2013 | Cassman et al. (2003b) |
| US-PFa | MF | Dwb | 1995 | 2014 | Yi et al. (2001) |
| US-Prr | ENF | Dwc | 2010 | 2014 | Ruehr et al. (2012d) |
| US-SP1 | ENF | BWh | 2000 | 2005 | Thomas et al. (1999a) |
| US-SP2 | ENF | Csa | 1998 | 2004 | Thomas et al. (1999b) |
| US-SP3 | ENF | Csa | 1999 | 2004 | Thomas et al. (1999c) |
| US-SRG | GRA | BSh | 2008 | 2014 | Ruehr et al. (2012e) |
| US-SRM | WSA | BSh | 2004 | 2014 | Scott et al. (2008) |
| US-Syv | MF | Dfb | 2001 | 2014 | Desai et al. (2005) |
| US-Ton | WSA | Csa | 2001 | 2014 | Tang et al. (2003) |
| US-Twt | CRO | Csb | 2009 | 2014 | Hatala et al. (2012) |
| US-UMB | DBF | Dfb | 2000 | 2014 | Rothstein et al. (2000) |
| US-UMd | DBF | BWk | 2007 | 2014 | Nave et al. (2011) |
| US-Var | GRA | Csa | 2000 | 2014 | Xu et al. (2004) |
| US-WCr | DBF | Dfb | 1999 | 2014 | Potter et al. (2001) |
| US-Whs | OSH | BWk | 2007 | 2014 | Scott et al. (2006) |
| US-Wkg | GRA | BWk | 2004 | 2014 | Emmerich (2003) |
| ZA-Kru | SAV | BSh | 2000 | 2013 | Archibald et al. (2009) |
| ZM-Mon | DBF | Aw | 2000 | 2009 | Merbold et al. (2009) |

**Table B1.** PCMCI parameters that were used differently from default settings.

| PCMCI parameter | Setting |
|---|---|
| significance $\alpha$ | 0.1 |
| $\alpha_{pc}$ | None |
| tau_min | 0 |
| tau_max | 5 |
| mask_type | 'y' |
| fdr_method | 'fdr_bh' |
| selected_links<br>(for variable set [$R_g$, T, NEE, VPD, H, LE]) | {0: [],<br>for i in [1,2,3,4,5]:<br>i:[(i,-1), (i,-2)] + [(j,0), (j,-1), (j,-2) for j in [1,2,3,4,5] and j$\neq$i]} |

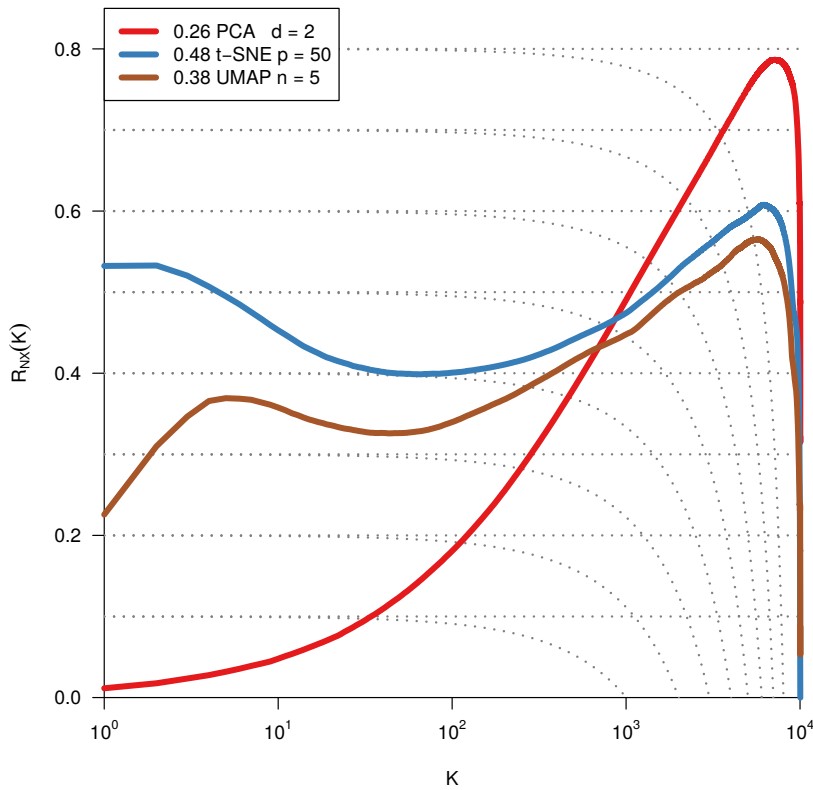

**Figure A1.** Quality assessment of dimensionality reduction techniques. To visualize and subsequently analyse the network space we reduce its dimensionality. We compared PCA, t-SNE and UMAP including various parameter settings (here: PCA's leading two principal components, t-SNE with perplexity 30, and UMAP with $n_{neighbors}$ equal 5 for 2 dimensions). The test statistic $R_{NX}(k)$ (y-axis) gives the improvement of the embedding of $k$-neighborhoods (x-axis) over a random embedding. The area under the curves (preserving the log-scaled x-axis) is given in the legend and gives an idea of the overall quality of the embedding Lee et al. (2015). We chose t-SNE with perplexity 30, as it preserves best local neighbourhoods and performs well on larger distances.

**Appendix C: Results and Discussion**

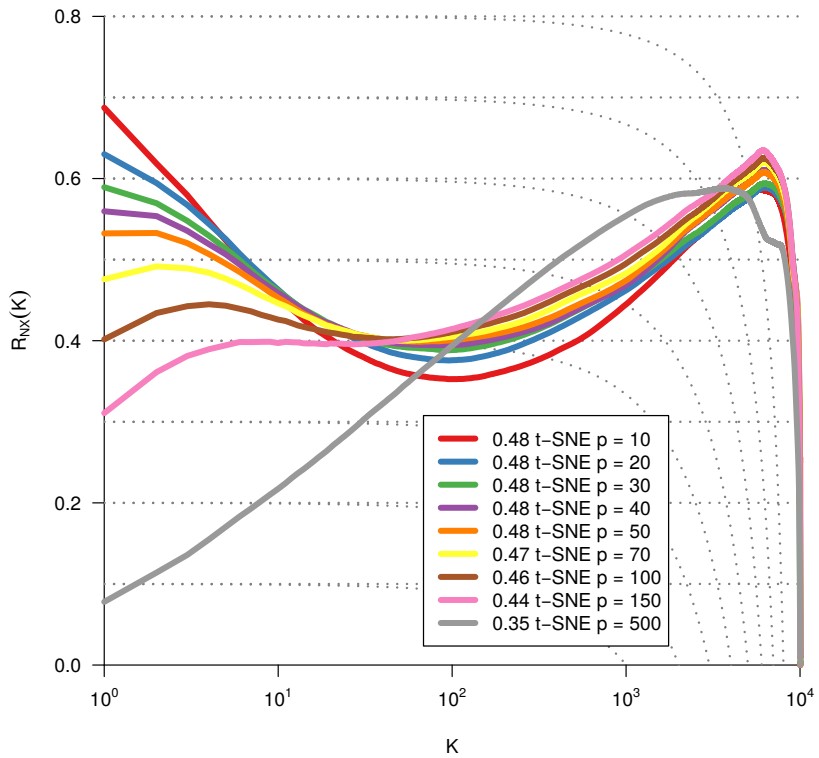

**Figure B1.** Same metric as Fig. A1. Optimisation of the dimensionality reduction via t-SNE by using different perplexity values.

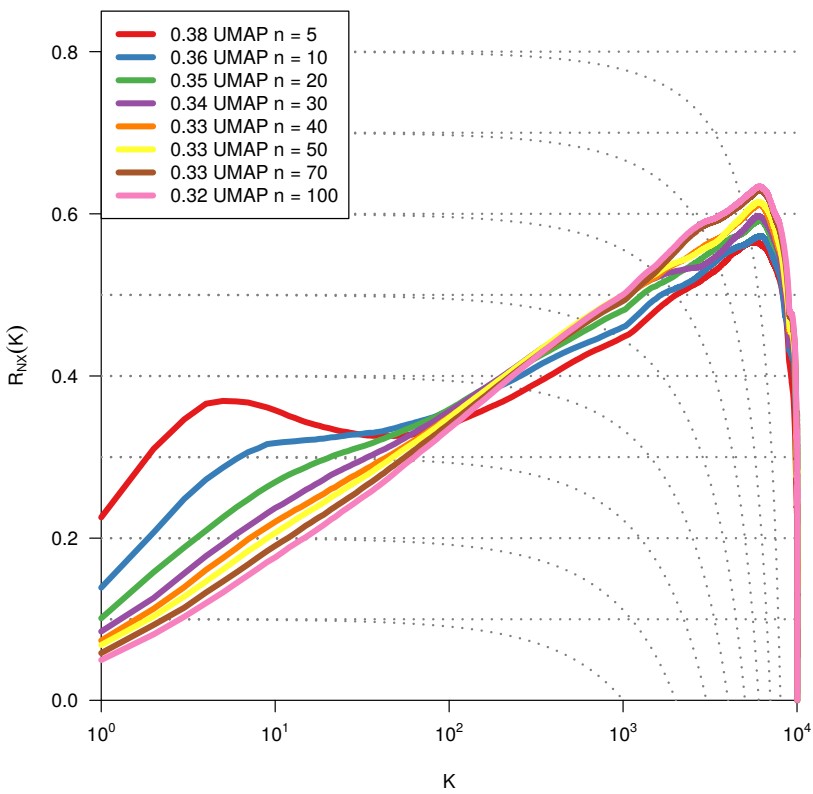

**Figure C1.** Same metric as Fig. A1. Optimisation of the dimensionality reduction to two dimensions via UMAP by using different values for the parameter $n_{neighbors}$.

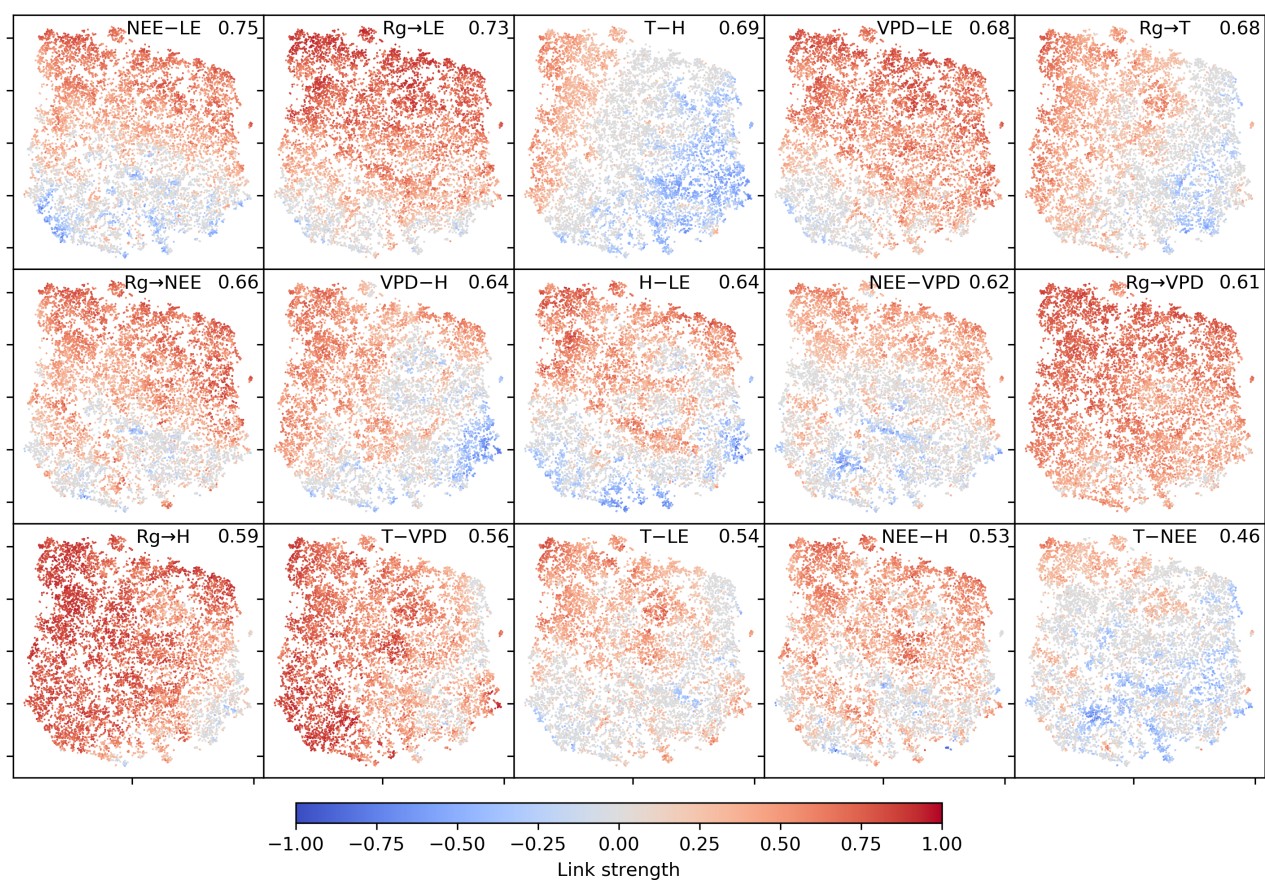

**Figure D1.** As Fig. 2 but produced from a new run of the analysis which does allow influences on $R_g$. The orientation of data points in this plot has changed compared to Fig. 2 due to the stochastic nature of t-SNE. But the embedding is almost merely mirrored.

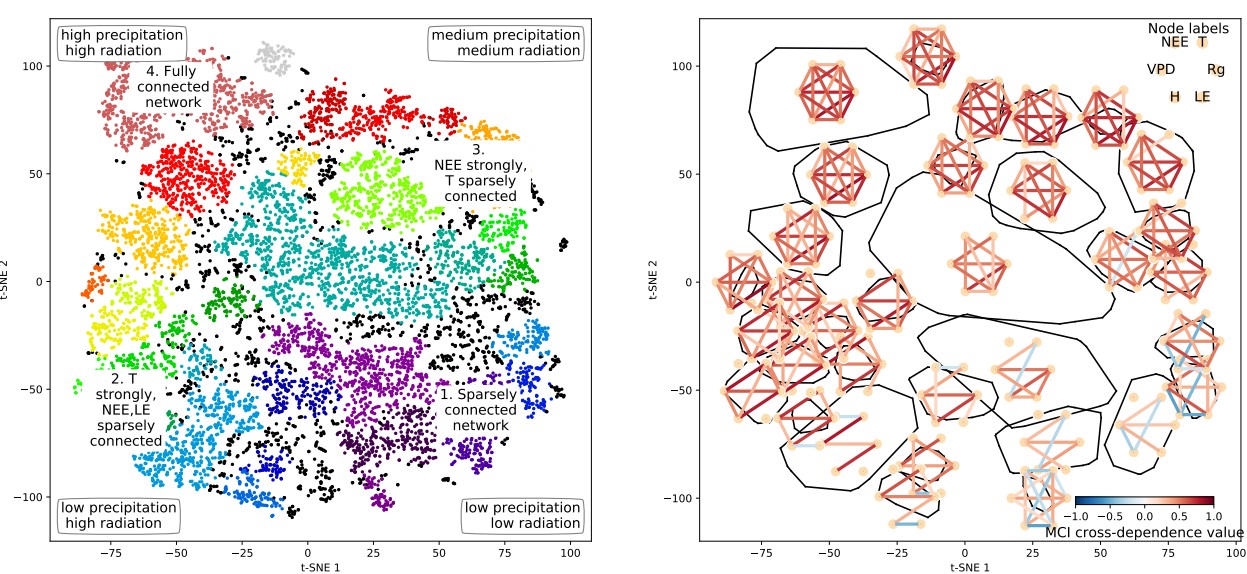

**Figure E1.** As Fig. 4 but with smaller clusters exhibiting the finer structure of the t-SNE space.

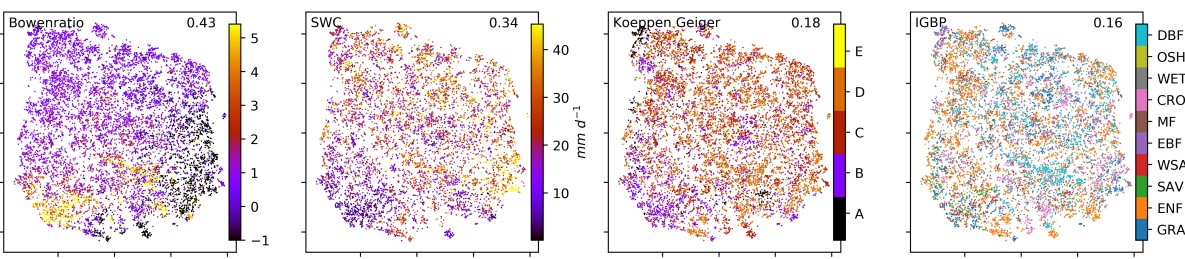

**Figure E2.** t-SNE space coloured by underlying mean Bowenratio and precipitation, as well as the ecosystems respective Koeppen Geiger class and IGBP type.

*Author contributions.* CK and MDM designed the study with contributions from all other authors. CK conducted the analysis and wrote the manuscript. All authors helped to improve the manuscript.

*Competing interests.* The authors declare that they have no competing financial interests.

*Acknowledgements.* D.G.M. acknowledges support from the European Research Council under grant agreement number 715254 (DRY-2-DRY). CK thanks the Max Planck Research School for global Biogeochemical Cycles for supporting his PhD project as well as Jacob A. Nelson in helping to assemble the dataset. This work used eddy covariance data acquired and shared by the FLUXNET community, including these networks: AmeriFlux, AfriFlux, AsiaFlux, CarboAfrica, CarboEuropeIP, CarboItaly, CarboMont, ChinaFlux, Fluxnet-Canada, Green-Grass, ICOS, KoFlux, LBA, NECC, OzFlux-TERN, TCOS-Siberia, and USCCC. The ERA-Interim reanalysis data are provided by ECMWF and processed by LSCE. The FLUXNET eddy covariance data processing and harmonization was carried out by the European Fluxes Database Cluster, AmeriFlux Management Project, and Fluxdata project of FLUXNET, with the support of CDIAC and ICOS Ecosystem Thematic Center, and the OzFlux, ChinaFlux and AsiaFlux offices.

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
