# Peer review of "Functional convergence of biosphere–atmosphere interactions in response to meteorological conditions"

_Biogeosciences, 2020_

## Referee Comment (RC1) · Anonymous Referee #1 · 27 Oct 2020

Review of Biogeosciences 2020-374

Title: Functional convergence of biosphere-atmosphere interactions in response to meteorology

General Comments:

The manuscript, "Functional convergence of biosphere-atmosphere interactions in response to meteorology," investigates a number of variables and their connections from publicly-available FLUXNET datasets using a relatively novel causal analysis method called "Peter Clark Momentary Conditional Independence" (PCMCI), in conjunction with a dimensionality reduction technique called "t-distributed stochastic neighbor embedding" (t-SNE) and a subsequent clustering algorithm called "Ordering Points To

Identify the Clustering Structure" (OPTICS). The specific research questions motivating the study are not clearly stated; the general motivation provided is, "to investigate how biosphere-atmosphere interactions vary across vegetation types and climate zones." This the manuscript accomplishes through a notion of linkages between biosphere and atmospheric variables, with the primary units of analysis being 1) network representations of those variables and their causal interactions over three-month windows at daily scale, 2) a two-dimensional representation of the structure of those high-dimensional networks, and 3) clusters formed within that 2d space of high dimensional networks.

The methods will likely be unfamiliar to most readers, and the units of analysis are quite abstract and require considerable explanation for readers to fully grasp the results being presented: this explanation is not currently sufficient in the manuscript.

Broad discussion focuses on some very interesting topics, such as 1) the universality or functional convergence of biosphere/atmosphere processes, 2) trajectories of ecosystems through a 2d space of land surface "network" states, including seasonal cycles and deviations due to extreme events, and 3) linkages between biosphere and atmospheric variables, and how their causal relationships could be represented as clinal processes along some continuum from linked to unlinked. Ultimately though, this discussion turns back to separating water/energy/radiation/temperature limitations on ecosystem productivity from land-atmosphere feedbacks (both of which are areas of deep physical research), which leads the reader to ask what the analysis gains from combining them in the first place.

While providing an interesting lens for looking at highly complex interactions between the biosphere and atmosphere across time and space, I found that this study failed to specify its intents and rather motivated too much using the tools (which instead should be motivated as useful for answering the question at hand). This led to results for which I am hard-pressed to find applications. I am not convinced that sufficiently substantial conclusions have been reached. I recommend a major structural overhaul of the paper, driven by specific, answerable scientific questions. At the same time–and

this is the difficulty in a study with such "boutique" methodology (with no judgement passed on that label)–the readers will still need \*more\* description of what is being shown in the analysis, all leading back to the primary research questions. I have tried to provide as specific guidance as I can in the following comments.

Specific Comments: What is/are the primary research question(s) being asked here? What is the knowledge gap?

The fundamental units of analysis need to be very clearly specified, as readers will be unfamiliar. Each point in Figure 1 is a network of connections between a bunch of variables at daily scale, but each representing three months of data, including some lagged effects. This network is the primary unit of analysis. It would really help to show an example of one of these networks at one location for three months before jumping into Figure 1, even though the authors have written papers on these networks before.

The authors need to discuss seasonality at some point earlier in the analysis to let readers know that all seasons will be studied, and that points in Figure 1 will represent different locations and different times of year. Line 80: (Probably an easy, but major point) "A comprehensive description from theoretical assumptions. . ." These assumptions should be stated clearly here, as the method is not well-known. As with any paper using basic regression analysis, a statement of the ways in which the analysis meets basic methodological assumptions is necessary. Rationale/justification for using the method when assumptions are not met are necessary as well. Some of this discussion can happen in supplementary materials if it is particularly involved, but the assumptions and their validity should probably be stated in the main text.

Line 96: "Unobserved common drivers can still render links as spurious." How do the critical \*non-stationary\* variables (at the time scale of your analyses) of biomass and phenostage influence 1) the validity of the estimation of your networks, and 2) the structure of the 2d space in Figs 1 and 2?

Line 105: "subtracted a smoothed seasonal mean from each variable. . ." I agree that

this needs to be done to remove non-stationarity which can cause spurious correlations. At the same time, subtraction of a Fourier series from a time series could either solve that problem or partially solve the problem while introducing new non-stationarities. How robust is the de-seasoning technique? Do the results change if you use other filtering methods?

Use of PCMCI, t-SNE, and OPTICS really makes this difficult for readers to follow the methodology. I would guess almost no one (particularly outside the author's list) is familiar with all of these. The authors need to motivate why they are using these methods with respect to some research question, and not just because causal tools exist.

Line 156: "The strongest gradients measured via distance correlations..." As the manuscript stands, I don't think even the most careful methodologically-focused readers are going to know how to interpret these results. It took me a lot of re-reading to get the idea that the distance correlations represent the spatial (in this 2d space) coherence of the link strengths. The link strengths themselves should be more clearly explained and motivated, probably in a preliminary figure showing an example network. The meaning of the link strength should be clarified (does link strength 1 mean fully causal? Completely dependent? One-to-one?)

Figure 1: As the manuscript stands, I do not think readers are equipped to understand what is being shown in this figure, which needs to change. While some methods may be dense and opaque, results need to be comprehensible to readers in the field, even if they are not close enough to the sub-field method specifics.

While Szekely et al. 2007 is highly-cited in the statistics literature, it is unlikely that many of your readers in biogeosciences will be familiar. How do we interpret these distance correlations? Having referred to Szekely myself, I can see that the correlations are metrics of dependence between random vectors, but can you clarify what are the vectors in question here (say for NEE-LE)? What is their dimensionality, what are the

constituent dimensional components? Are they across space and time (I think so) and season of the year (I don't think so), and if so, how do these constituent components combine to give a single number (rho=0.75)? Does this represent something like a fraction of explained variance, and if so across what conditions? Can I compare the rho for NEE-LE and the rho for T-H to infer something about bivariate coupling? What time-scale should I think about these metrics representing? Mostly daily? Does Rg-> H mean that Rg almost always causes H (with positive partial correlation)? Does T-VPD being red mean that T causes VPD with positive correlation or that VPD causes T with negative correlation? Your readers need their hands held through all of this to interpret your results and see the patterns you are seeing in your analysis.

Figure 1 caption: "As Rg can only be a cause. . ." Is this true? I'd imagine that LE -> Rg often if Rg is measured at the tower (as opposed to top of atmosphere). There are certainly LE -> humidity -> cloud formation processes at the local scale in many locations, aren't there?

Line 156: "The colouring reveals that the link strengths are ordered along gradients." This sounds like A Finding, but of course the world is spatially autocorrelated and neighborhoods are similar. What is it explicitly that is interesting about this? Is it expected or unexpected (I would expect it) and why?

Figure 2: This figure could in theory be used to add interpretability to Figure 1, but otherwise the main take-away is that GPP, NEE, and LE are fairly correlated, as are Rg and T. I enjoy looking at this for patterns, and I can imagine spending time in a study looking for structure and emergent relationships in this data projection, but I am not sure what "results" it represents. How could I as a biogeoscientist make use of the information in this Figure? What question is this helping to answer?

Line 165: "The results show that a high dimensional space encompassing more than 10000 ecosystem networks representing the states of biosphere–atmosphere interactions from ecosystems of various geographic origins can be reduced to a compact twodimensional manifold characterized by four edges and gradients of biosphere and atmosphere conditions." Maybe I'm missing a key piece of nuance here. It is by definition true that applying a dimensionality-reduction algorithm to high-dimensional data will yield a lower dimensional representation. Are you claiming the positive (and sufficiently large absolute values) of the distance correlation metric imply something more significant about biosphere-atmosphere interactions and coupling? Isn't the t-SNE method designed to do something *close* to maximizing these distance correlations? And didn't you select your dimensionality reduction method to basically do that (maybe not with the explicit cost function of maximal distance correlations, but with local and global neighborhood coherence maximization)? I'm either 1) not seeing how this is a finding rather than the necessary outcome of your approach, or 2) not seeing how significant these specific metrics are relative to what I should be expecting (maybe the rho values would for some reason be expected to all be less than 0.1 for some reason?). I think it is well-known that there are continua of all of these variables (ranges of GPP, ranges of LE, etc.) and that stepping from one location to another nearby location (in space, time, or say VPD space) will lead to small changes in the biospheric and atmospheric states. This is not surprising. If you can quantify or qualify something ABOUT those gradients, it would be very interesting because that science is wide-open, but I don't see how Figure 2 is doing that. I see that you are suggesting that this is not obvious in the statement, "While gradients in MCI partial correlation strength are expected as they were used as features in the dimensionality reduction, gradients in climatic and biospheric conditions were not." But I don't see that as actually surprising—there are entire disciplines focused on the biogeographical structure of ecosystems and their gradients. How could we not expect a clinal change in LE and GPP to be related to a clinal change in LE-GPP coupling?

Line 183: "LE and NEE are weakly, not, or even negatively connected" This is interesting because it is commonly thought that arid/semi-arid locations have the highest coupling between LE (or Bowen ratio) and NEE because of omnipresent water limitation (e.g., in references below). Are these networks so arid as to not have vegetation?

Dirmeyer, P. A., F. J. Zeng, A. Ducharne, J. C. Morrill, and R. D. Koster, 2000: The Sensitivity of Surface Fluxes to Soil Water Content in Three Land Surface Schemes. J. Hydrometeor., 1, 121–134, https://doi.org/10.1175/1525-7541(2000)001<0121:TSOSFT>2.0.CO;2. Williams, I. N., Lu, Y., Kueppers, L. M., Riley, W. J., Biraud, S. C., Bagley, J. E., and Torn, M. S. (2016), Land–atmosphere coupling and climate prediction over the U.S. Southern Great Plains, J. Geophys. Res. Atmos., 121, 12,125– 12,144, doi:10.1002/2016JD025223. Short Gianotti, D. J., Rigden, A. J., Salvucci, G. D., & Entekhabi, D. (2019). Satellite and station observations demonstrate water availability's effect on continental–scale evaporative and photosynthetic land surface dynamics. Water Resources Research, 55, 540– 554. https://doi.org/10.1029/2018WR023726. Crow, W. T., and Coauthors, 2020: Soil Moisture–Evapotranspiration Overcoupling and L-Band Brightness Temperature Assimilation: Sources and Forecast Implications. J. Hydrometeor., 21, 2359–2374, https://doi.org/10.1175/JHM-D-20-0088.1.

Figure 3: How were these archetypal clusters determined, by eye? It's a little weird to define 17 clusters algorithmically and then define 4 clusters of clusters by hand. Do the 4 types fall out on their own if restricted to 4 clusters? Type 2 looks like barren, arid landscapes. Type 3 is the mid-latitudes growing season. Type 1 is winter. Type 4 is interesting in that I wouldn't expect strong coupling in the tropics between T and anything or NEE and anything, since coupling (and even causality) is generally thought of as related to bottleneck variables. What leads to this full connectedness in physical terms do you think?

Line 149: "The monthly median network is the *average* of the networks..." The mixing of average and median here is complicating an already complex processing step. Are these averages being taken in the 2d reduced space axes?

Line 216: "for a given month" One month or three month window, shifting by one month at a time? This is confusing throughout. If using overlapping data (single-month network definition, but using sliding three-month windows), I think that will cause some

real problems in discussions of the inter-connectedness of your neighborhoods. Your rho values in Figs 1 and 2 will be artificially high by triply counting your data. I am sure you don't want this, but I think you need to either switch to 1-month windows or remove any networks with overlapping data windows (which will reduce your data points by a factor of 3 if I am understanding the method correctly). You can't talk about how nice and smooth the 2d space is when the analysis units are all 2/3 the same data as other neighboring units.

Figure 4: Why aren't Bowen ratios defined for the whole year for any of the sites except US-SRM? Aren't those variable observed? Don't you need them to map the seasonal trajectories for the plot on the left? Maybe not, and a lot of the network points are fit with partial data? Is that a problem in terms of robustness of the 2d space, the clusters, or the link strengths? You need to clarify how you deal with missing data throughout.

Lines 229-230: The fact that you are post-hoc trying to talk about this in terms of water/energy/temperature limitation on ecosystem productivity, but then calling out a separate, loosely connected concept of atmospheric interaction covariance highlights a general weakness in your storyline. There are physical concepts that are well-understood here: water, energy, and radiation (and temperature) can all act as limitation on photosynthetic activity in ways we largely understand (at the plot scale). At the same time, the land surface and atmosphere feed back on one another. I respect and am intrigued by the way in which you are attempting to link those two, but the question remains: what are we learning about how the land surface works by doing so? This is a major issue to be resolved for this manuscript.

Figure 5: I like the idea of this figure, and think it is a compelling way to look at extreme events. At the same time, it is worth asking whether this 2d space is good at representing extreme events. Does it make sense to think that a tropical rainforest undergoing an extreme drought is really just suddenly (and temporarily) turning into a system akin to a woody savanna, with all of the accompanying *causal* land-atmosphere feedbacks and carbon-energy-water coupling? I wouldn't assume so. That doesn't mean that this isn't

a fine first-order way to think about extreme events from a new analytical framework, but I would not a priori think that this is physically representative in any way beyond very coarse correlational descriptions. Presumably extreme events are another suite of dimensions that could be characterized, except for their lack of statistical representation in any data set (by definition). This warrants an explanation and some discussion of limitations.

Technical Corrections:

Line 29: "only consider two variables. . ." Granger Causality and Transfer Entropy at this point are only reasonably considered bivariate if you state "bivariate Granger Causality" as the authors do. This bivariance by necessity stance is a false position to take here. I don't know as much about CCM (although a quick search turns up a few recent multivariate extensions), but no econometricians think of VAR-based GC as bivariate, and there have been wikipedia articles about multivariate mutual information for more than a decade. I don't know that you need to have this discussion, depending on how you re-frame your research questions and motivation, but you can't publish this sort of claim.

Line 38-42: A nice synthesizing motivation. The motivation for the tie-in to *extremes* is not very clear though. Are you going to be looking at just biosphere-atmosphere interactions under meteorologically-extreme conditions? Or across the whole range of observed conditions?

Line 59: Strange citation format for Nelson.

Line 87-95: In the partial correlations, are the correlations controlling for (multiple) lags of the X and Y variables as well, or just other variables?

Line 116: And SWC?

Line 147: "non-intercepting convex hulls. . ." Even as a very methodological reader I am completely lost here. Is this a typo? What does non-intercepting mean? Intersecting

maybe?

Line 180: "Leave" -> leaf

Line 226: month-> months

---

## Referee Comment (RC2) · Anonymous Referee #1 · 28 Oct 2020

I wanted to add also that I can see the tremendous, deep work that went into this analysis, presumably with a lot of thought about how to visualize highly complex physical/spatial/temporal processes. I do think that the analysis has real promise, and the figures (once understood by the new reader) convey a huge amount of information at once. That said, all of that work and promise goes unrealized without a compelling question motivating the narrative. I think the real work to be done now is in the manuscript, and much less in the methods. With that motivating narrative I think this work could be a very insightful paper.

My apologies for not highlighting the real potential in this work more clearly in the original review comment.

-Reviewer 1

---

## Referee Comment (RC3) · Anonymous Referee #2 · 6 Nov 2020

This is a very interesting paper on biosphere-atmosphere interaction, but it is also a bit difficult for me to understand. I have some background in causality detection (Granger Causality, CCM), but still find the paper hard to follow when I read it for the first time. This is mostly due to the incomplete description of the methods the authors used. When I went back and read the paper the same lead author published earlier this year (Kirch et al 2020, BG), this paper becomes clearer.

The author used a causal relationship detection method, PCMCI, and quantify the interactions between biosphere and atmosphere (represented by the energy, water and carbon fluxes measured by eddy covariance flux towers). With the resultant 10038 networks obtained from PCMCI, the authors applied a dimension reduction algorithm and visualize and analyze these networks along two dimensions. By quantifying the

2-dimension space into four regions, the authors can show the trajectory of biosphere atmosphere interactions changes through time and across different sites. The authors lastly claim that environment are the major factors that regulates the biosphere-atmosphere interactions, effect from the vegetation type is small.

This paper tackles a very important question, and use rather novel method. And also because of this, the presentation is not very clear. The final conclusion is drawn based on qualitative evidence which weakens the importance of this study. I have several comments below for the authors to consider.

1. From a reader perspective, I find this paper difficult to read. Clearly, there is a large gap between "what the readers know" and "what the authors assume the readers know". For example, in the introduction (P2 L33) when the author first mention PCMCI, I would expect further explanation on this new method because it is clearly not known by most or at least some readers. Instead, the author did not give any explanation on this but just mentioned one paper. I think this is a good place where the authors can briefly explain what is the basic ideas behind this method and what kind of information it can tell us. With this in mind, the readers can better follow the research question. Another example is in section 2.5, the author mentioned a method called OPTICS, but also did not provide enough explanations. There are also some good practices to improve the readability. For example, in the last paragraph of the introduction where the authors describe the structure of the paper. The authors can define the aim of each section first before directly stating what they did in each section. This also apply for the method description which is very dense and filled with lots of jargons and acronyms.

2. The major conclusion of this study, to me, is not well supported. The authors claim that the biosphere-atmosphere interactions are determined mostly by the environment, with limited effect from vegetation types, etc. This is supported by the similar causal network shown in the 2-D space of the t-SNE. However, there are two problems with this. First, when looking at this network, much of the linkage are physically based, with limited effect of vegetation, for example, the relationship between T-VPD, T-H, Rg-

T, etc. Vegetation would have limited effect on these relationships and for the other carbon fluxes related relationships, different biome types may have different linkage strength. For example, in Figure 1, when looking at link strength related to NEE and other variable, the patterns become more sporadic, especially at bottom left corner, these differences in responses may be caused by the vegetation types or differences in months, but may have limited contribution to the overall network. That is to say, the vegetation types can have effect on the biosphere-atmosphere interaction, but only contribute limited to the network evaluated, therefore, their effects are ignored. Another problem related to this is that the authors showed several cases of the change in climate can cause a shift in interaction in the t-SNE space. However, these effects are not quantitively analyzed, how much of this change in interaction strength are caused by climate and how much is caused by differences in ecosystems. are they both significant enough? Without these information, the conclusion is draw without solid support.

Some detailed comments: P2L49, "one high dimension observation" is not clear. Is it better to say "one facet in the high dimension space"

P4L92, based on the information theory, the causal relationship that happens within the smallest time step of observation cannot be detected. For example, although we know that Rg has a causal relationship with NEE, but this happens in seconds or minutes considering the lags in measurements, this causal relationship cannot be detected by the algorithm and will be regarded as unidirectional. This need to be mentioned or discussed as a limitation for interpretation of the results.

P5L129, This is not clear enough, is computation efficiency the only difference? Why would it generate different results as compared to t-SNE?

P7 Fig 1, Rg can also be affected by the cloud which can be affected by ET, H and other factors. See Green et al. 2017 Nature Geoscience.

P11 Fig 4, Maj->May

---

## Referee Comment (RC4) · Anonymous Referee #3 · 10 Nov 2020

This paper aims to show that biosphere-atmosphere interactions are driven by meteorological conditions, and that these meteorological conditions produce similar biosphere-atmosphere interactions, regardless of climate gradient and ecosystem type. It concludes that based on these results, similar principles can be used to serve as empirical references for global vegetation models regardless of region and ecosystem type. This study uses observational FLUXNET data, combined with a novel causal method known as PCMCI.

Should these results be true, I believe that the findings have the potential to be quite helpful to the modeling community—perhaps we can stop trying to incorporate so many individual processes that vary between regions, and instead streamline the process using the knowledge of this shared behavior. However, there is not a lot of information on the methods, making a reader feel that either one must simply 'trust' the results, or go on one's own intense literature search to try and understand the paper. This is not a well-known methodology, and thus it is even more important than usual to go above and beyond convincing the reader that this is a sound and reliable method. While including other papers as references to the method is great, a reader should not be required to track these down to make sense of the paper.

Additionally, the figures that accompany the methods are non-intuitive and need better description– once again, making the paper, including the results section, quite difficult to follow.

Minor comments: Line 28: Change from 'they allow to infer' to 'they allow one to infer' Line 88: MCI has been previously defined Line 100-102 Remove the word 'well' L114-What characteristics impinge on performance? This isn't explained. Also, why use precipitation as binary (rain/no rain) rather than a time series? Why can't precipitation be used in the model? Line 143- Add 'The' before following Line 164- Insert 'as' into 'Are not as much' Table A1—This table would be better with more information. Potentially adding ecosystem type/climate zone/type determined by PCMCI analysis of each flux tower site would be useful. Table B1—Shouldn't radiation be added as a variable?

---

## Author Comment (AC2) · 18 Nov 2020

**This is a very interesting paper on biosphere-atmosphere interaction, but it is also a bit difficult for me to understand. I have some background in causality detection (GrangerCausality, CCM), but still find the paper hard to follow when I read it for the first time. This is mostly due to the incomplete description of the methods the authors used. When I went back and read the paper the same lead author published earlier this year (Kirch et al 2020, BG), this paper becomes clearer. The author used a causal relationship detection method, PCMCI, and quantify the interactions between biosphere and atmosphere (represented by the energy, water and carbon fluxes measured by eddy covariance flux towers). With the resultant 10038 networks obtained from PCMCI, the authors applied a dimen-**

sion reduction algorithm and visualize and analyze these networks along two dimensions. By quantifying the 2-dimension space into four regions, the authors can show the trajectory of biosphere–atmosphere interactions changes through time and across different sites. The authors lastly claim that environment are the major factors that regulates the biosphere-atmosphere interactions, effect from the vegetation type is small. This paper tackles a very important question, and use rather novel method. And also because of this, the presentation is not very clear. The final conclusion is drawn based on qualitative evidence which weakens the importance of this study. I have several comments below for the authors to consider.

We thank the reviewer for the effort undertaken to give us valuable feedback to improve the manuscript. We will attempt to improve mentioned weaknesses and consider suggested improvements.

1. **From a reader perspective, I find this paper difficult to read. Clearly, there is a large gap between "what the readers know" and "what the authors assume the readers know". For example, in the introduction (P2 L33) when the author first mention PCMCI, I would expect further explanation on this new method because it is clearly not known by most or at least some readers. Instead, the author did not give any explanation on this but just mentioned one paper. I think this is a good place where the authors can briefly explain what is the basic ideas behind this method and what kind of information it can tell us. With this in mind, the readers can better follow the research question. Another example is in section 2.5, the author mentioned a method called OPTICS, but also did not provide enough explanations. There are also some good practices to improve the readability. For example, in the last paragraph of the introduction where the authors describe the structure of the paper. The authors can define the aim of each section first before directly stating what they did in each section. This also apply for the**

**method description which is very dense and filled with lots of jargons and acronyms.**

We can comprehend the reviewers critique on sparsely given information regarding the applied methods (PCMCI P2L33, Optics Sect. 2.5). The brevity though is not due to some sort of sloppiness. We tried to provide as much information as necessary to gain an intuition of the methods while reading the main text. For example, based on the paragraph (L27:L36) the reader could gain following understanding about PCMCI: PCMCI is a causal discovery method, that allows to create causal networks based on a set of assumptions. It is a multivariate approach attempting to remove the influence of third variables when evaluation the dependence among two variables. The method was already tested on datasets similar to the ones considered in the present study. In short, what is the method supposed to do, how is it achieved and does it work for the intended use? Yet, we fully respect the reviewers opinion, are grateful that it is stated and acknowledge the effect of "what the readers know" and "what the authors assume the readers know". When revising the manuscript we will focus on increasing the level of information. This will go in hand with addressing suggestions of reviewer 1. We further appreciate the suggestion how to improve the description of the structure of the paper.

2. **The major conclusion of this study, to me, is not well supported. The authors claim that the biosphere-atmosphere interactions are determined mostly by the environment, with limited effect from vegetation types, etc. This is supported by the similar causal network shown in the 2-D space of the t-SNE. However, there are two problems with this. First, when looking at this network, much of the linkage are physically based, with limited effect of vegetation, for example, the relationship between T-VPD, T-H, Rg-T, etc. Vegetation would have limited effect on these relationships and for the other carbon fluxes related relationships, different biome types may have**

**different linkage strength. For example, in Figure 1, when looking at link strength related to NEE and other variable, the patterns become more sporadic, especially at bottom left corner, these differences in responses may be caused by the vegetation types or differences in months, but may have limited contribution to the overall network. That is to say, the vegetation types can have effect on the biosphere-atmosphere interaction, but only contribute limited to the network evaluated, therefore, their effects are ignored. ...**

The reviewer is right with the statement 'much of the linkage are physically based, with limited effect of vegetation, for example, the relationship between T-VPD, T-H, Rg-T, etc'. However, it seems to be overlooked that we addressed this in L255:259. Four (if including H) variables are physical/atmospheric variables, two are biospheric variables. The most dominant links (Figure 1) though include the biospheric variables. Further Figure 4 shows, that the transitions between the archetypes are dominated by changes of biosphere variable dependencies. In addition, we show the distribution of IGBP classes in Figure D2 which shows a much lower distance correlation with the axes than any three month mean value of a physical variable. The reason for overlooking this argumentation might be the fact that the role of the distance correlation as well as the information revealed by the dimensionality reduction step are not clearly posed (as reviewer 1 suggests). We therefore currently see this comment addressed with measures undertaken to address comments of reviewer 1.

3. **... Another problem related to this is that the authors showed several cases of the change in climate can cause a shift in interaction in the t-SNE space. However, these effects are not quantitively analyzed, how much of this change in interaction strength are caused by climate and how much is caused by differences in ecosystems. are they both significant enough? Without these information, the conclusion is draw without solid support.**

The reviewer is presumably talking about Figure 4 and 5. Figure 4 shows median trajectories (kind of mean seasonal cycle) in the t-SNE space for five ecosystems. In addition to the median trajectories, we show the corresponding mean values of radiation, precipitation and Bowen ratio. The choice of these variables is made because Figure 2 shows that transitions between archetype one and four covary with energy availability (Rg, T) and transitions between archetype two and three covary with water availability (P, SWC, Bowenratio). Figure 5 shows strong deviations (due to changes in interaction strength) from such median trajectories for certain ecosystems. The strong deviations in the trajectory coincide with strong anomalies in precipitation (also shown).

The suggested comparison of 'how much of this change in interaction strength are caused by climate and how much is caused by differences in ecosystems' thus proves difficult or trivial, as we look at trajectory changes within one ecosystem (Figure 5). Thus the ecosystem stays constant resulting in a covariance of zero with any given variable. Other ecosystem variables like phenological changes within that ecosystem are again driven by climate. Figure 4 indeed allows a comparison of ecosystems and actually is intended to do so. Figure 2 shows, that the low dimensional space covaries with atmospheric variables. This would cause ecosystems with prevailing atmospheric conditions to populate certain regions in that low dimensional space. Figure 4 shows that this is the case. We choose the ecosystems due to their contrasting climate. Further ecosystem effect quantification again appears trivial. The only ecosystem effect quantification that appeared sensible to us within the scope of the manuscript is to quantify the covariance of the low dimensional embedding with the IGBP class, i.e. the covariance of IGBP class underlying each network with its location in the low dimensional space. This is done using the metric distance correlation and the result is shown in Figure D2.

As we discussed at the end of the manuscript (L260ff), we as well regard it im-
portant to quantify the effect of biotic factors on the network structure. However, this would include the use of further datasets as for instance standage, vegetation coverage, soil properties and species diversity. Additionally, the framework would need to be developed further to quantify and compare changes in the t-SNE space. We regard it to be out of scope of this paper.

4. **Some detailed comments: P2L49, "one high dimension observation" is not clear. Is it better to say "one facet in the high dimension space"** Thanks for pointing out a lack of clarity. Yet, we do not see that the suggestion improves the situation into the correct direction. We suggest: 'Each of the estimated networks constitutes one observation, i.e. measurement, in a high-dimensional space. This space is spanned by the network's links.'

5. **P4L92, based on the information theory, the causal relationship that happens within the smallest time step of observation cannot be detected. For example, although we know that Rg has a causal relationship with NEE, but this happens in seconds or minutes considering the lags in measurements, this causal relationship cannot be detected by the algorithm and will be regarded as unidirectional. This need to be mentioned or discussed as a limitation for interpretation of the results.**

We agree with the reviewer. NEE, i.e. carbon uptake into the biosphere due to photosynthesis, responses much quicker to Rg than the time resolution of 30 minutes would allow to detect. This leads to non-directed (not unidirectional) links. Such links are called contemporaneous links in PCMCI (no direction of influence can be inferred). This can be indeed more clearly stated. Actually, this is going to happen when stating the assumptions underlying PCMCI, as reviewer 1 suggested (see comment 4).

6. **P5L129, This is not clear enough, is computation efficiency the only difference? Why would it generate different results as compared to t-SNE?**

Indeed, the given differences between t-SNE and UMAP are a bit shallow. We add a bit more description.

7. **P7 Fig 1, Rg can also be affected by the cloud which can be affected by ET, H and other factors. See Green et al. 2017 Nature Geoscience.** Also reviewer one pointed to this possibility. We acknowledge the possibility of Rg being affected by sensible (H) or latent (LE) heat fluxes which is investigated in Green et al. 2017 Nature Geoscience. However, here satellite data is used, not eddy covariance data. The larger the area over which the variables are aggregated, the higher the possibility to detect any effect of LE or H on Rg. Thus, we regard the possibility of Le and H influencing Rg and its effect as rather small as we look on ecosystem level. We therefore decided to do without detecting this influence in favour of setting Rg as a main driver.

Setting Rg as a main driver will avoid the estimation of any driver of Rg an thus any regression on Rg. This has the benefit, that any seasonal changes in the variables (i.e. remaining non-stationarities) caused by seasonal changes in radiation are attributed to radiation.

8. **P11 Fig 4, Maj→May** Thanks.

---

## Author Response (AR1)

**Response to reviewer 1**

The manuscript, "Functional convergence of biosphere-atmosphere interactions in response to meteorology," investigates a number of variables and their connections from publicly-available FLUXNET datasets using a relatively novel causal analysis method called "Peter Clark Momentary Conditional Independence" (PCMCI), in conjunction with a dimensionality reduction technique called "t-distributed stochastic neighbor embedding" (t-SNE) and a subsequent clustering algorithm called "Ordering Points To Identify the Clustering Structure" (OPTICS). The specific research questions motivating the study are not clearly stated; the general motivation provided is, "to investigate how biosphere–atmosphere interactions vary across vegetation types and climate zones." This the manuscript accomplishes through a notion of linkages between biosphere and atmospheric variables, with the primary units of analysis being 1) network representations of those variables and their causal interactions over three-month windows at daily scale, 2) a two-dimensional representation of the structure of those high-dimensional networks, and 3) clusters formed within that 2d space of high dimensional networks. The methods will likely be unfamiliar to most readers, and the units of analysis are quite abstract and require considerable explanation for readers to fully grasp the results being presented: this explanation is not currently sufficient in the manuscript. Broad discussion focuses on some very interesting topics, such as 1) the universality or functional convergence of biosphere/atmosphere processes, 2) trajectories of ecosystems through a 2d space of land surface "network" states, including seasonal cycles and deviations due to extreme events, and 3) linkages between biosphere and atmospheric variables, and how their causal relationships could be represented as clinal processes along some continuum from linked to unlinked. Ultimately though, this discussion turns back to separating water/energy/radiation/temperature limitations on ecosystem productivity from land-atmosphere feedbacks (both of which are areas of deep physical research), which leads the reader to ask what the analysis gains from combining them in the first place. While providing an interesting lens for looking at highly complex interactions between the biosphere and atmosphere across time and space, I found that this study failed to specify its intents and rather motivated too much using the tools (which instead should be motivated as useful for answering the question at hand). This led to results for which I am hard-pressed to find applications. I am not convinced that sufficiently substantial conclusions have been reached. I recommend a major structural overhaul of the paper, driven by specific, answerable scientific questions. At the same time – and this is the difficulty in a study with such "boutique" methodology (with no judgement passed on that label)– the readers will still need *more* description of what is being shown in the analysis, all leading back to the primary research questions. I have tried to provide as specific guidance as I can in the following comments.

We thank the reviewer for this comprehensive review and the critical view

on our manuscript. We recognise the potential to improve the accessibility of the manuscript. The main adjustments include the revision of the introduction as well as an added subsection to the method section. Further changes have been done throughout the text.

**Specific comments:**

1. **What is/are the primary research question(s) being asked here? What is the knowledge gap?**

   We indeed did not formulate a specific research question as we conducted a rather exploratory study which was motivated in line 37ff. The world is attempted to be categorised: ecosystems by their appearance, or climate regions by their temperature and precipitation. While discrete categories contradict the natural continuum, they hold certain benefits. Here we wanted to examine, whether ecosystems show distinct functional states and how these functional states can be characterised. Do those states form a continuum or separate classes? Are ecosystems limited within the accessible states? The special quality of our approach is that the states characterise interactions and only those, i.e they are not build by the mean of a certain variable but are an aggregation of many. To straighten our story line, we added following hypothesis to our introduction:

   We hypothesise first that the accessible states of biosphere–atmosphere interactions are limited and can be characterised by few functional states despite the complexity and differences among ecosystems. Second, attributing to an ecosystems adaptation, we further hypothesise that specific ecosystem can only access a limited fraction of the functional states.

2. **The fundamental units of analysis need to be very clearly specified, as readers will be unfamiliar. Each point in Figure 1 is a network of connections between a bunch of variables at daily scale, but each representing three months of data, including some lagged effects. This network is the primary unit of analysis. It would really help to show an example of one of these networks at one location for three months before jumping into Figure 1, even though the authors have written papers on these networks before.**

   Thanks for reminding us that our daily work isn't that of others and that we need to guide readers more carefully. We regard it as sensible to address this comment together with comment 4. We added a schematic as Figure 1 which supports the explanation of our work flow which is again explained in an added paragarph at the end of the method section.

3. **The authors need to discuss seasonality at some point earlier in the analysis to let readers know that all seasons will be studied, and that points in Figure 1 will represent different locations and different times of year.**

   see comment 2. Figure 1 now includes a one year excerpt of of some time series. Further we mention, that we work on daily time resolution and calculate networks for each month in the subsection 'Network Estimation'.

4. **Line 80: (Probably an easy, but major point) "A comprehensive description from theoretical assumptions..." These assumptions should be stated clearly here, as the method is not well-known. As with any paper using basic regression analysis, a statement of the ways in which the analysis meets basic methodological assumptions is necessary. Rationale/justification for using the method when assumptions are not met are necessary as well. Some of this discussion can happen in supplementary materials if it is particularly involved, but the assumptions and their validity should probably be stated in the main text.**

We agree, 'easy but major'. Thus we listed the assumptions in the method subsection PCMCI and explained to which degree they are accounted for and which effects that has in the Subsection 'Network Estimation'.

5. **Line 96: "Unobserved common drivers can still render links as spurious." How do the critical \*non-stationary\* variables (at the time scale of your analyses) of biomass and phenostage influence 1) the validity of the estimation of your networks, and 2) the structure of the 2d space in Figs 1 and 2?**

The reviewer points to one assumption out of the set of assumptions requested in comment 5. Causal stationarity is an assumption of PCMCI. This does not mean that a causal dependence may not change in strength but that it persists over the time period of interest. Causal stationarity would not be given for many ecosystems when estimating networks over a whole year. We attempt to increase causal stationarity by chunking the time series into 3 month periods. Especially in autumn or spring one can still argue causal stationarity to be violated but as we are talking of a gradual shift, the network representation remains a valid representation of the functional state. An example can be given by rather consistent trajectories in Figure 4 and 5.

We see the comment addressed by the answer to comment 5).

6. **Line 105: "subtracted a smoothed seasonal mean from each variable..." I agree that this needs to be done to remove non-stationarity which can cause spurious correlations. At the same time, subtraction of a Fourier series from a time series could either solve that problem or partially solve the problem while introducing new non-stationarities. How robust is the de-seasoning technique? Do the results change if you use other filtering methods?**

In a previous paper (Krich et al. 2020) we studied the performance of PCMCI on an artificial dataset. Here we could show, that the subtraction of the seasonal mean by a Fourier Series does decrease the false positive rate but leaves the true positive rate mostly unaffected.

We had referenced to Krich et al. 2020 already. We added the explanation: 'This decreases the detection of false links while leaving the detection of true links largely unaffected.'

7. **Use of PCMCI, t-SNE, and OPTICS really makes this difficult for readers to follow the methodology. I would guess almost no**

one (particularly outside the author's list) is familiar with all of these. The authors need to motivate why they are using these methods with respect to some research question, and not just because causal tools exist.

We agree, that the combination of methods is heavy but it is indeed mandatory to address our research question (see answer to comment 1). For example, in Krich et al. 2020 we could show, that PCMCI enables to focus on a few but relevant links compared to correlation. Due to the quantity of observed ecosystems and the high dimensionality we require further methods for the analysis. We will motivate their use more strongly and clarify their purpose. It is actually not needed to understand each method to its details.

We met the demand by adding a new subsection (Sect. Workflow) and the schematic Fig. 1 to the methods. Additionally paragraph two of the introduction is altered to better address the motivation to use PCMCI.

8. **Line 156: "The strongest gradients measured via distance correlations..." As the manuscript stands, I don't think even the most careful methodologically-focused readers are going to know how to interpret these results. It took me a lot of rereading to get the idea that the distance correlations represent the spatial (in this 2d space) coherence of the link strengths. The link strengths themselves should be more clearly explained and motivated, probably in a preliminary figure showing an example network. The meaning of the link strength should be clarified (does link strength 1 mean fully causal? Completely dependent? One-to-one?)**

We gave a short explanation of distance correlation in a new subsection of the methods. Additionally we clarified its purpose and use in the new Sect. Workflow. The link strength is now visualised in Fig 1 c) and explained in the method subsection PCMCI more clearly: ' the MCI value gives an estimate of dependence between two time series, one potentially lagged, with the influence of other lagged drivers including autocorrelation removed, yielding a better interpretation of the strength of a causal mechanism than the common Pearson correlation. For a more detailed discussion of the interpretation, see Runge2019a. As a particular partial correlation, the MCI value is independent of the variables' mean value and is normalised in [-1, 1] and can, hence, be compared between variable pairs with different units of measurement. '

9. **Figure 1: As the manuscript stands, I do not think readers are equipped to understand what is being shown in this figure, which needs to change. While some methods maybe dense and opaque, results need to be comprehensible to readers in the field, even if they are not close enough to the sub-field method specifics.**

Thanks for pointing out this issue. Having read the newly added subsection Workflow and the Fig. 1, the previous Figure 1 , now Fig. 2, should be better understandable.

10. **While Szekely et al. 2007 is highly-cited in the statistics litera-ture, it is unlikely that many of your readers in biogeosciences will be familiar. How do we interpret these distance correla-tions? Having referred to Szekely myself, I can see that the correlations are metrics of dependence between random vectors, but can you clarify what are the vectors in question here (say for NEE-LE)? What is their dimensionality, what are the con-stituent dimensional components? Are they across space and time (I think so) and season of the year (I don't think so), and if so, how do these constituent components combine to give a single number (rho=0.75)? Does this represent something like a fraction of explained variance, and if so across what condi-tions? Can I compare the rho for NEE-LE and the rho for T-H to infer something about bivariate coupling? What time-scale should I think about these metrics representing? Mostly daily? Does Rg$\rightarrow$ H mean that Rg almost always causes H (with pos-itive partial correlation)? Does T-VPD being red mean that T causes VPD with positive correlation or that VPD causes T with negative correlation? Your readers need their hands held through all of this to interpret your results and see the patterns you are seeing in your analysis.**

The performed and already previously mentioned actions shall be suffi-cient to address the questions and remove uncertainties: '... The domi-nant features are the links that appear with strong gradients in the low dimensional embedding. To quantify and later rank the gradients exhib-ited by each link, we use the measure distance correlation (see Sect. **??**). Therefore, we calculate the distance correlation of the link strengths (1d) with their position on the low dimensional embedding axes (2d). ' and whole Subsection 'Distance Correlation'.

11. **Figure 1 caption: "As Rg can only be a cause..." Is this true? I'd imagine that LE $\rightarrow$Rg often if Rg is measured at the tower (as opposed to top of atmosphere). There are certainly LE $\rightarrow$ humidity $\rightarrow$ cloud formation processes at the local scale in many locations, aren't there?**

We have as well considered the possibility for such processes. We have come to the conclusion, that LE can affect Rg but likely does so at an other location (due to lateral transport). Thus we decided to set Rg as the main driver of our system.

To not exclude the possibility of variables influencing Rg, we changed the sentence to: 'As $R_g$ is set as potential driver (PCMCI parameter 'se-lected_links', see table **??**), connections including $R_g$ are directed $\rightarrow$.' And further added to the Subsection Network estimation: ' We acknowledge the possibility of $R_g$ being influenced by other variables, e.g. via transpi-ration and subsequent cloud formation. Yet, on the ecosystem scale we work with, we presume this effect to be rather small and likely dominated by lateral transport. Besides these possibilities, setting $R_g$ as driver can account for remaining non stationarities Runge2018.'

12. **Line 156: "The colouring reveals that the link strengths are ordered along gradients." This sounds like A Finding, but of course the world is spatially autocorrelated and neighborhoods are similar. What is it explicitly that is interesting about this? Is it expected or unexpected (I would expect it) and why?**

We indeed see this as a minor (not major!) result/finding even though we as well expected it (as we wrote in line 168). Yet, we do not see spatial autocorrelation as the reason. First of all, we are dealing with data on ecosystem level and on this spatial scale FLUXNET towers can be regarded as rather sparsely distributed ( in contrast to e.g. satellite data). Second, Figure 1 displays the distribution of link strengths, i.e. the strength of interaction between a set of variables resembling atmosphere and biosphere. No information of location enters here and as can be seen in Fig. 1, the information of location exhibit a lower distance correlation with the tSNE axis as any link.

That Figure 1 of the manuscript is presented as a finding builds upon following: By attempting to preserve local neighbourhoods, tSNE also preserves gradients within the data. The stronger the gradient the more likely it is to observe it in the low dimensional embedding. Within our approach, only link strengths are handed to tSNE. If the networks are not random (which we expect), we will see gradients of link strength . Further, even though gradients in link strength are expected, we did not know which are dominating before doing the analysis. Figure 1 presents the emerging gradients ranked via distance correlation.

These facts should be clarified with the added subsections, which for example state: 'Projecting this high dimensional space onto two dimensions (Fig. **??**e) allows first of all for visualisation. In case the data consists of a structure that can be 'identified' by the dimensionality reduction method, the visualisation reveals the dominant features of transitions between different states of biosphere–atmosphere interactions. The dominant features are the links that appear with strong gradients in the low dimensional embedding. To quantify and later rank the gradients exhibited by each link, we use the measure distance correlation (see Sect. **??**).' (Sect. workflow). or ' This procedure makes t-SNE very good at visualising clusters in the data and non-linear relationships.' (Sect. Dimensionality Reduction).

13. **Figure 2: This figure could in theory be used to add interpretability to Figure 1, but otherwise the main take-away is that GPP, NEE, and LE are fairly correlated, as are Rg and T. I enjoy looking at this for patterns, and I can imagine spending time in a study looking for structure and emergent relationships in this data projection, but I am not sure what "results" it represents. How could I as a biogeoscientist make use of the information in this Figure? What question is this helping to answer?**

As stated already in the answer to comment 12, only link strength values enter the dimensionality reduction process. Thus, any gradient that emerges besides those of link strength values is unexpected. Figure 2 in the manuscript shows 3 month mean values. As the value range of variables is not per se affecting any link strength (see Fig. 2, the finding that

[Figure]

Figure 1: Same as Figure 2 of the manuscript but colored by latitude and longitude. The distance correlation value of (upper right of each inset) is much lower than that of any link. This indicates that location and spatial autocorrelation can not be used to explain the gradients of the link strength.

certain links correlate with a mean variable value is interesting. For example, such correlation could help to tailor the dependence structure in model parametrisations.

We expect the value of Figure 2 (now Fig. 3) to become clearer with the measures explained already in the answer of comment 12 and the sentences: ' We also examine the distance correlation of secondary quantities with the axes. The secondary quantities are firstly mean values of variables calculated for each three month period of network estimation as well as secondly static values like climate class, vegetation type or location. The secondary quantities are used to find covariates of the low dimensional embedding that can help to explain its structure. (Sect. Workflow).

14. **Line 165: "The results show that a high dimensional space encompassing more than 10000 ecosystem networks representing the states of biosphere–atmosphere interactions from ecosystems of various geographic origins can be reduced to a compact two-dimensional manifold characterized by four edges and gradients of biosphere and atmosphere conditions." Maybe I'm missing a key piece of nuance here. It is by definition true that applying a dimensionality-reduction algorithm to high-dimensional data will yield a lower dimensional representation.**

The reviewer is right, a dimensionality-reduction algorithm will project any high dimensional data onto a low dimensional space, i.e. 2 dimensions. However, it is not granted, that the projection yields any meaningful insight. PCA for instance projects the high dimensional network space onto the axes with the highest explained variances. However, as Figure A1 shows, points which have been far away in the high dimensional space are now lying close to each other. t-SNE on the contrary manages to unwrap the intrinsically high dimensional network space. Further, as gradients are preserved according to their strength, the embedding also reveals which links dominate transitions between the networks (Figure 1 of the manuscript). Further, the space has a rather quadrilateral shape (if we would not use any significance threshold, that shape would be more prominent). Having four corners is neither anything that is known beforehand.

**Are you claiming the positive (and sufficiently large absolute values) of the distance correlation metric imply something more significant about biosphere-atmosphere interactions and coupling?** no **Isn't the t-SNE method designed to do something \*close\* to maximizing these distance correlations?** yes **And didn't you select your dimensionality reduction method to basically do that (maybe not with the explicit cost function of maximal distance correlations, but with local and global neighborhood coherence maximization)?** yes **I'm either 1) not seeing how this is a finding rather than the necessary outcome of your approach, or 2) not seeing how significant these specific metrics are relative to what I should be expecting (maybe the rho values would for some reason be expected to all be less than 0.1 for some reason?).**

Again, many of the already stated changes address these questions: Among others: 'Projecting this high dimensional space onto two dimensions (Fig. ??e) allows first of all for visualisation. In case the data consists of a structure that can be 'identified' by the dimensionality reduction method, the visualisation reveals the dominant features of transitions between different states of biosphere–atmosphere interactions.'

**I think it is well-known that there are continua of all of these variables (ranges of GPP, ranges of LE, etc.) and that stepping from one location to another nearby location (in space, time, or say VPD space) will lead to small changes in the biospheric and atmospheric states. This is not surprising. If you can quantify or qualify something ABOUT those gradients, it would be very interesting because that science is wide-open, but I don't see how Figure 2 is doing that. I see that you are suggesting that this is not obvious in the statement, "While gradients in MCI partial correlation strength are expected as they were used as features in the dimensionality reduction, gradients in climatic and biospheric conditions were not." But I don't see that as actually surprising as there are entire disciplines focused on the biogeographical structure of ecosystems and their gradients. How could we not expect a clinal change in LE and GPP to be related to a clinal change in LE-GPP coupling?**

Whether there are continua or not and independent of how variable values among ecosystems change, a link strength estimated via PCMCI is not dependent on the mean value of its variables. To visualise that, we make use of an artificial dataset created from random variables with preset dependencies (Fig. 2 left column). Scaling this dataset by a factor (we chose 10) (Fig. 2 right column) PCMCI detects the same network (including link strength). This is due to the fact that the independence test partial correlation, before assessing any dependence, standardises the data.

We added the sentence ' As a particular partial correlation, the MCI value is independent of the variables' mean value and is normalised in [-1, 1] and can, hence, be compared between variable pairs with different units of measurement.' to the PCMCI section.

Additionally, the contemporaneous link NEE–LE, for example, changes

[Figure]

Figure 2: We created a dataset with given dependencies from random variables. The difference between the left and the right column is simply that the dataset in multiplied by 10 (see y axis). The network detected by PCMCI is the same for both datasets.

not necessarily according to changes in position (latitude, longitude) as Fig. 4 demonstrates.

15. **Line 183: "LE and NEE are weakly, not, or even negatively connected" This is interesting because it is commonly thought that arid/semi-arid locations have the highest coupling between LE (or Bowen ratio) and NEE because of omnipresent water limitation (e.g., in references below). Are these networks so arid as to not have vegetation? [1, 2, 3, 4]**

I guess here we have a misunderstanding due to some imprecise wording. The sentence is changed to: LE and NEE are weakly, not, or even negatively connected to the atmosphere.

But still, the statement is true also for the connection between NEE and LE. The ecosystems showing such states are vegetated. Yet, depending on the state, this vegetation might be dormant or even dead (grass cover).

We adapted a sentence to 'Low water availability but high temperatures cause ecosystems to enter a dormant state which leads to low carbon and water fluxes and low connectivity.'

16. **Figure 3: How were these archetypal clusters determined, by eye? It's a little weird to define 17 clusters algorithmically and then define 4 clusters of clusters by hand. Do the 4 types fall out on their own if restricted to 4 clusters? Type 2 looks like barren,arid landscapes. Type 3 is the mid-latitudes growing season. Type 1 is winter. Type4 is interesting in that I wouldn't expect strong coupling in the tropics between T and anything or NEE and anything, since coupling (and even causality) is generally thought of as related to bottleneck variables. What leads to this full connectedness in physical terms do you think?**

[Figure]

Figure 3: Behaviour of link strength NEE–LE in the climate space precipitation - temperature.

The low dimensional embedding takes the form of quadrilateral shape (the lower the applied significance threshold, the more prominent the four corners, the less prominent the clusters). The four archetypes are the average networks found in the clusters at each corner of the low dimensional embedding (line 288). Defining such archetypes is based on the concept of endmember states. We try to clarify the choice of the archetypes. The full connectedness can be explained the following way. We focus on the example of T and NEE: The optimum temperature range for photosynthesis is between 10 and 20 °C. This temperature range is given in archetype 3. As any fluctuation in this temperature range barely affects photosynthesis (it remains close to optimum), T and NEE are unconnected. In archetype 4 the temperature is above 20°C and thus affecting photosynthesis which links T and NEE. Similarly, radiation changes can be detected in photosynthetic activity linking Rg and NEE. High water availability and energy input allows for large latent heat fluxes and stomata to remain open linking Rg and LE as well as NEE and LE.

To clarify the identification of the archetypes, we added: 'This visualisation reveals that the mean networks of the clusters situated at the embedding's edges can be regarded as archetypes of network structures, i.e. extremal, characteristic states (similar to the concept of endmember states). The four states can be described as follows:'

17. **Line 149: "The monthly median network is the \*average\* of the networks..." The mixing of average and median here is complicating an already complex processing step. Are these averages being taken in the 2d reduced space axes?**

Indeed, using both terms can be confusing but their use is intended. The 'median networks' are calculated using a concept of a median calculation in 2d, which is why we call them 'median' instead of 'mean' networks. Unfortunately, their calculation involves the calculation of an average (similar to the median calculation in 1d with an even number of observations).

We regard the choice of words justified to maintain mathematical correctness.

18. **Line 216: "for a given month" One month or three month window, shifting by one month at a time? This is confusing throughout. If using overlapping data (single-month network definition, but using sliding three-month windows), I think that will cause some real problems in discussions of the inter-connectedness of your neighborhoods. Your rho values in Figs 1 and 2 will be artificially high by triply counting your data. I am sure you don't want this, but I think you need to either switch to 1-month windows or remove any networks with overlapping data windows (which will reduce your data points by a factor of 3 if I am understanding the method correctly). You can't talk about how nice and smooth the 2d space is when the analysis units are all 2/3 the same data as other neighbouring units**

Each month of each year is attributed a network. This network is calculated from a time period of three month (centred three month window).

Therefore, the reviewer is right when saying that (almost) each datapoint is used three fold. This might also increase the distance correlations calculated for Figure 1 and 2 compared to taking only every third month (every third network). However, this does not affect the result. As the distance correlation is only used to rank the links. If 'triply counting' increases the distance correlation value, it is done for all links alike. We do not calculate networks on one month time windows as this would be too few data points. Further, for example the median networks of the month February and May of the towers DE-Hai and FI-Hyy lie pretty far apart. Including March and April thus appears mandatory to create the trajectories.

Based on our reasoning and that the purpose of distance correlation is clarified we kept the figure and analysis unchanged.

19. **Figure 4: Why aren't Bowen ratios defined for the whole year for any of the sites except US-SRM? Aren't those variable observed? Don't you need them to map the seasonal trajectories for the plot on the left? Maybe not, and a lot of the network points are fit with partial data? Is that a problem in terms of robustness of the 2d space, the clusters, or the link strengths? You need to clarify how you deal with missing data throughout.**

Thanks for pointing out this inaccuracy. Bowen ratios are defined for each month but since we use a log scale, we can not plot them when negative. Setting a log scale also might not be necessary. Thus the graph could be changed to Fig. 4. Nevertheless, neither of the values on the right of Fig 4 is needed to map/plot the trajectories on the left. The trajectories are based on the network structure (link strengths) only. Missing datapoints are flagged and not included in any calculation.

We changed the figure and added the sentence ' In winter month the Bowen ratio can turn negative. Nevertheless we set the lower limit of the y-axis to 0.' tu the caption. Further we added 'Missing data was flagged as such and is ignored by PCMCI.' to the subsection Sect. Network Estimation.

20. **Lines 229-230: The fact that you are post-hoc trying to talk about this in terms of water/energy/temperature limitation on ecosystem productivity, but then calling out a separate, loosely connected concept of atmospheric interaction covariance highlights a general weakness in your storyline. There are physical concepts that are well-understood here: water, energy, and radiation (and temperature) can all act as limitation on photosynthetic activity in ways we largely understand (at the plot scale). At the same time, the land surface and atmosphere feed back on one another. I respect and am intrigued by the way in which you are attempting to link those two, but the question remains: what are we learning about how the land surface works by doing so? This is a major issue to be resolved for this manuscript.**

We hope to understand this comment correctly. We are not using our analysis to post-hoc identify limitations of productivity. Instead, we are trying to understand how biosphere–atmosphere interactions vary: Which states of interactions exist? Which interaction states are dominating?

[Figure]

Figure 4:

When are the different states reached?. This is very different from: What are the limitations of GPP? Yet, the way we refer to Kraemer et al. 2020 might cause the impression. When rewriting the manuscript, we will more clearly link the finding to the actual research question and clarify how findings of Kraemer et al. 2020 support our findings.

We removed the reference to Kraemer et al. from this part and instead added: 'These behaviours demonstrate what the previous figures (Fig. **??** and **??**) have already suggested: Ecosystem's populate the low dimensional space and migrate within as allowed by their climatic conditions. Thereby they can exhibit a wide range of interaction structures as can be seen from the mid-latitude sites. As these behaviours are multi year averages they could resemble more ecosystem adaptation to median climatic conditions than flexible adjustment of biosphere–atmosphere interactions to quickly changing meteorological conditions. If biosphere–atmosphere interactions are confined by adaptation shall be investigated in the final analysis section.'

21. **Figure 5: I like the idea of this figure, and think it is a compelling way to look at extreme events. At the same time, it is worth asking whether this 2d space is good at representing extreme events. Does it make sense to think that a tropical rainforest undergoing an extreme drought is really just suddenly (and temporarily) turning into a system akin to a woody savanna, with all of the accompanying \*causal\* land-atmosphere feedbacks and carbon-energy-water coupling? I wouldn't assume so. That doesn't mean that this isn't a fine first-order way to think about extreme events from a new analytical framework, but I would not a priori think that this is physically representative in any way beyond very coarse correlational descriptions. Presumably extreme events are another suite of dimensions that could be characterized, except for their lack of statistical rep-**

**resentation in any data set (by definition). This warrants an explanation and some discussion of limitations.**

A tropical rainforest will not turn suddenly into a woody savanna (structure wise). The processes we capture in our interaction networks are certainly not covering all processes that characterise ecosystem and distinguish them from each other. However, our analysis shows, that the biosphere-atmosphere interactions (limited to the chosen set of variables within the analysis) can become very similar.

We added a subsection to discuss limitations: '

**0.1  Limitations of the study**

Finally, we would like to take a critical view on our analysis approach. As stated in Sect. **??**, PCMCI might fail to identify some spurious links due to the occurrence of contemporaneous confounders. Thus networks can not be interpreted causally but this does not severely hinder their value for the current analysis. In addition we include a rather limited set of variables into the network estimation. Thus we cannot and do not claim that ecosystems become fully alike under similar meteorological conditions. Yet, on the timescale investigated the data shows, that the interactions among the chosen set of variables can be described by very similar structures. Follow up studies might search for and include further biosphere variables. Currently, an analysis of the biotic effects on the network structure is hampered because the t-SNE space is not metric. Thus, for instance, the effect of a drought with similar magnitude in a boreal and temperate forest cannot simply be compared by the deviation from their median trajectory.'

22. **Technical Corrections:**

23. **Line 29: "only consider two variables..." Granger Causality and Transfer Entropy at this point are only reasonably considered bivariate if you state "bivariate Granger Causality" as the authors do. This bivariance by necessity stance is a false position to take here. I don't know as much about CCM (although a quick search turns up a few recent multivariate extensions), but no econometricians think of VAR-based GC as bivariate, and there have been wikipedia articles about multivariate mutual information for more than a decade. I don't know that you need to have this discussion, depending on how you reframe your research questions and motivation, but you can't publish this sort of claim.**

The reviewer is right in stating that multivariate extensions exists. Though their practicality for large datasets (especially transfer entropy) is low, as of high computational complexity and data size requirements.

We accounted for the reviewers concern by changing the reasoning of our introduction (second paragraph).

24. **Line 38-42: A nice synthesizing motivation. The motivation for the tie-in to \*extremes\* is not very clear though. Are you going to be looking at just biosphere–atmosphere interactions under meteorologically-extreme conditions? Or across the whole range of observed conditions?**

   We agree with the reviewer. Linking the motivation to extremes in not needed. We removed it and added the hypothesis instead.

25. **Line 59: Strange citation format for Nelson.** Thanks, we changed the format.

26. **Line 87-95: In the partial correlations, are the correlations controlling for (multiple) lags of the X and Y variables as well, or just other variables?**

   Before calculating partial correlations between Y at time t and X at time t-tau (tau between 0 and 5) other correlating variables, i.e. drivers, are removed via regression. Those drivers can be third variables Z(t-tau) with tau from 1 to 5 and the past of Y(t), Y(t-tau) with tau from 1 to 5 as well as the past of X(t-tau) which is given for tau+1 to 5.

   We added the sentence: 'The conditions Z can consist of third variables or the past of X and Y'

27. **Line 116: And SWC?** We added the explanation: 'The issue with SWC is its lower availability and for those sites that have such measurements it might be applied at differing depth. The depth that is mostly present is at shallows depth of 5 or 10 cm. The upper soil layer, however, dries out quickly and can explain only little of the latent heat flux.'

28. **Line 147: "non-intercepting convex hulls..." Even as a very methodological reader I am completely lost here. Is this a typo? What does non-intercepting mean? Intersecting maybe?**

   The reviewer is right. We changed the phrase to 'non-intersecting convex hulls'.

29. **Line 180: "Leave" → leaf**

30. **Line 226: month→ months** We will perform the suggested changes.

**Response to reviewer 2**

**This is a very interesting paper on biosphere-atmosphere interaction, but it is also a bit difficult for me to understand. I have some background in causality detection (GrangerCausality, CCM), but still find the paper hard to follow when I read it for the first time. This is mostly due to the incomplete description of the methods the authors used. When I went back and read the paper the same lead author published earlier this year (Kirch et al 2020, BG), this paper becomes clearer. The author used a causal relationship detection method, PCMCI, and quantify the interactions between biosphere and atmosphere (represented by the energy, water and carbon fluxes**

measured by eddy covariance flux towers). With the resultant 10038 networks obtained from PCMCI, the authors applied a dimension reduction algorithm and visualize and analyze these networks along two dimensions. By quantifying the 2-dimension space into four regions, the authors can show the trajectory of biosphere–atmosphere interactions changes through time and across different sites. The authors lastly claim that environment are the major factors that regulates the biosphere-atmosphere interactions, effect from the vegetation type is small. This paper tackles a very important question, and use rather novel method. And also because of this, the presentation is not very clear. The final conclusion is drawn based on qualitative evidence which weakens the importance of this study. I have several comments below for the authors to consider.**

We thank the reviewer for the effort undertaken to give us valuable feedback to improve the manuscript. We attempted to improve mentioned weaknesses by adding a new paragraph to the method section and restructuring the introduction.

1. **From a reader perspective, I find this paper difficult to read. Clearly, there is a large gap between "what the readers know" and "what the authors assume the readers know". For example, in the introduction (P2 L33) when the author first mention PCMCI, I would expect further explanation on this new method because it is clearly not known by most or at least some readers. Instead, the author did not give any explanation on this but just mentioned one paper. I think this is a good place where the authors can briefly explain what is the basic ideas behind this method and what kind of information it can tell us. With this in mind, the readers can better follow the research question. Another example is in section 2.5, the author mentioned a method called OPTICS, but also did not provide enough explanations. There are also some good practices to improve the readability. For example, in the last paragraph of the introduction where the authors describe the structure of the paper. The authors can define the aim of each section first before directly stating what they did in each section. This also apply for the method description which is very dense and filled with lots of jargons and acronyms.**

As we changed the introduction based on comments of reviewer one the introduction to PCMCI now reads: ' One of that group is PCMCI Runge2019a, a causal graph discovery algorithm based on a combination of the PC algorithm (named after its inventors Peter and Clark Spirtes1991) and the Momentary Conditional Independence (MCI) test Runge2019a. By applying such tests, it becomes possible to account for common drivers and mediators which can cause two variables to correlate even though, no direct causal link exists between them. Then MCI partial correlations estimated by PCMCI yield a better interpretation of the strength of a causal mechanism than the common Pearson correlation. [?] tested PCMCI regarding its suitability for interpreting eddy covariance data. The method proved to be consistent despite the data's inherent noisy character and

was capable to extract well interpretable interaction structures. A causal interpretation of specific links, though, has to take into account potentially unmet assumptions.'

With the new Subsection 'Workflow' we also explained how the methods build upon each other and which purpose they fulfil. We aimed to clarify the method descriptions and only slightly increased the information content of the already existing subsections of the methods, as we do not want to overload the paper with technical details not necessary to understand the results.

2. **The major conclusion of this study, to me, is not well supported. The authors claim that the biosphere-atmosphere interactions are determined mostly by the environment, with limited effect from vegetation types, etc. This is supported by the similar causal network shown in the 2-D space of the t-SNE. However, there are two problems with this. First, when looking at this network, much of the linkage are physically based, with limited effect of vegetation, for example, the relationship between T-VPD, T-H, Rg-T, etc. Vegetation would have limited effect on these relationships and for the other carbon fluxes related relationships, different biome types may have different linkage strength. For example, in Figure 1, when looking at link strength related to NEE and other variable, the patterns become more sporadic, especially at bottom left corner, these differences in responses may be caused by the vegetation types or differences in months, but may have limited contribution to the overall network. That is to say, the vegetation types can have effect on the biosphere-atmosphere interaction, but only contribute limited to the network evaluated, therefore, their effects are ignored. ...**

The reviewer is right with the statement 'much of the linkage are physically based, with limited effect of vegetation, for example, the relationship between T-VPD, T-H, Rg-T, etc'. However, it seems to be overlooked that we addressed this in L255:259. Four (if including H) variables are physical/atmospheric variables, two are biospheric variables. The most dominant links (Figure 1) though include the biospheric variables. Further Figure 4 shows, that the transitions between the archetypes are dominated by changes of biosphere variable dependencies. In addition, we show the distribution of IGBP classes in Figure D2 which shows a much lower distance correlation with the axes than any three month mean value of a physical variable. The reason for overlooking this argumentation might be the fact that the role of the distance correlation as well as the information revealed by the dimensionality reduction step are not clearly posed (as reviewer 1 suggests). We therefore see this comment addressed with measures undertaken to address comments of reviewer 1.

3. **... Another problem related to this is that the authors showed several cases of the change in climate can cause a shift in interaction in the t-SNE space. However, these effects are not quantitively analyzed, how much of this change in interaction strength are caused by climate and how much is caused by differences in**

**ecosystems. are they both significant enough? Without these information, the conclusion is draw without solid support.**

The reviewer is presumably talking about Figure 4 and 5. Figure 4 shows median trajectories (kind of mean seasonal cycle) in the t-SNE space for five ecosystems. In addition to the median trajectories, we show the corresponding mean values of radiation, precipitation and Bowen ratio. The choice of these variables is made because Figure 2 shows that transitions between archetype one and four covary with energy availability (Rg, T) and transitions between archetype two and three covary with water availability (P, SWC, Bowenratio). Figure 5 shows strong deviations (due to changes in interaction strength) from such median trajectories for certain ecosystems. The strong deviations in the trajectory coincide with strong anomalies in precipitation (also shown).

The suggested comparison of 'how much of this change in interaction strength are caused by climate and how much is caused by differences in ecosystems' thus proves difficult or trivial, as we look at trajectory changes within one ecosystem (Figure 5). Thus the ecosystem stays constant resulting in a covariance of zero with any given variable. Other ecosystem variables like phenological changes within that ecosystem are again driven by climate. Figure 4 indeed allows a comparison of ecosystems and actually is intended to do so. Figure 2 shows, that the low dimensional space covaries with atmospheric variables. This would cause ecosystems with prevailing atmospheric conditions to populate certain regions in that low dimensional space. Figure 4 shows that this is the case. We choose the ecosystems due to their contrasting climate. Further ecosystem effect quantification again appears trivial. The only ecosystem effect quantification that appeared sensible to us within the scope of the manuscript is to quantify the covariance of the low dimensional embedding with the IGBP class, i.e. the covariance of IGBP class underlying each network with its location in the low dimensional space. This is done using the metric distance correlation and the result is shown in Figure D2.

As we discussed at the end of the manuscript (L260ff), we as well regard it important to quantify the effect of biotic factors on the network structure. However, this would include the use of further datasets as for instance standage, vegetation coverage, soil properties and species diversity. Additionally, the framework would need to be developed further to quantify and compare changes in the t-SNE space. We regard it to be out of scope of this paper.

We added this limitation to the discussion.

4. **Some detailed comments: P2L49, "one high dimension observation" is not clear. Is it better to say "one facet in the high dimension space"** Thanks for pointing out a lack of clarity. Yet, we do not see that the suggestion improves the situation into the correct direction. Therefore we changed it to: 'Each of the estimated networks constitutes one observation in a high dimensional space with a network's links spanning its axes (Fig. **??**d). '

5. **P4L92, based on the information theory, the causal relationship that happens within the smallest time step of observation**

**cannot be detected. For example, although we know that Rg has a causal relationship with NEE, but this happens in seconds or minutes considering the lags in measurements, this causal relationship cannot be detected by the algorithm and will be regarded as unidirectional. This need to be mentioned or discussed as a limitation for interpretation of the results.**

We agree with the reviewer. NEE, i.e. carbon uptake into the biosphere due to photosynthesis, responses much quicker to Rg than the time resolution of 30 minutes would allow to detect. This leads to non-directed (not unidirectional) links. Such links are called contemporaneous links in PCMCI (no direction of influence can be inferred). This can be indeed more clearly stated. We addressed this in the Subsection Workflow: 'The strongest and most consistent links are contemporaneous, indicating that interactions happen on time scales shorter than the time resolution. While lagged common drivers are excluded, contemporaneous links can still be spurious due to contemporaneous confounding (see Sect. **??**). Nevertheless, we focus our analysis on these 15 links, as they contain most information. '

6. **P5L129, This is not clear enough, is computation efficiency the only difference? Why would it generate different results as compared to t-SNE?**

Indeed, the given differences between t-SNE and UMAP are a bit shallow. Yet pointing out the differences can become very technical. We added information for t-SNE and pointed to a characteristic difference with UMAP: 'In contrast t-SNE aims to preserve local neighbourhoods. Therefore it calculates first similarity scores for each point pair using euclidean distances and Gaussian distributions. Subsequently it randomly projects the data onto the lower dimensional space and attempts to rearrange points in a way that the previously determined similarities are obtained. To assess the similarities in the low dimensional space, however, it uses a Student-t distribution. This helps to separate points which are also originally separated. This procedure makes t-SNE very good at visualising clusters in the data and non-linear relationships. Drawbacks are the difficult interpretability of the embedding axes due to the non-linear nature and its fairly long computation time for large datasets. Further, distances between far separated points and those belonging to different clusters in the embedding space are not (necessarily) comparable to the original distances. This is as t-SNE does not preserve both the global and local structure at the same time, which is attempted by UMAP. UMAP was developed as an improvement of t-SNE regarding structure preservation and results also in a shorter run time especially for higher dimensions. A comparison of t-SNE and UMAP is given in appendix C in McInnes2018.'

7. **P7 Fig 1, Rg can also be affected by the cloud which can be affected by ET, H and other factors. See Green et al. 2017 Nature Geoscience.**

Also reviewer one pointed to this possibility. We acknowledge the possibility of Rg being affected by sensible (H) or latent (LE) heat fluxes which is investigated in Green et al. 2017 Nature Geoscience. However, here

satellite data is used, not eddy covariance data. The larger the area over which the variables are aggregated, the higher the possibility to detect any effect of LE or H on Rg. Thus, we regard the possibility of Le and H influencing Rg and its effect as rather small as we look on ecosystem level. We therefore decided to do without detecting this influence in favour of setting Rg as a main driver.

Setting Rg as a main driver will avoid the estimation of any driver of Rg and thus any regression on Rg. This has the benefit, that any seasonal changes in the variables (i.e. remaining non-stationarities) caused by seasonal changes in radiation are attributed to radiation.

We added the reasoning to Sect. Network Estimation: 'Further we set $R_g$ as a potential driver of the system (by excluding its parents from the PCMCI parameter 'selected_links', see table **??**). We acknowledge the possibility of $R_g$ being influenced by other variables, e.g. via transpiration and subsequent cloud formation. Yet, on the ecosystem scale we work with, we presume this effect to be rather small and likely dominated by lateral transport. Besides these possibilities, setting $R_g$ as driver can account for remaining non stationarities Runge2018.'

8. **P11 Fig 4, Maj→May** Thanks.

**Response to reviewer 3**

**This paper aims to show that biosphere-atmosphere interactions are driven by meteorological conditions, and that these meteorological conditions produce similar biosphere-atmosphere interactions, regardless of climate gradient and ecosystem type. It concludes that based on these results, similar principles can be used to serve as empirical references for global vegetation models regardless of region and ecosystem type. This study uses observational FLUXNET data, combined with a novel causal method known as PCMCI. Should these results be true, I believe that the findings have the potential to be quite helpful to the modeling community and perhaps we can stop trying to incorporate so many individual processes that vary between regions, and instead streamline the process using the knowledge of this shared behavior. However, there is not a lot of information on the methods, making a reader feel that either one must simply 'trust' the results, or go on one's own intense literature search to try and understand the paper. This is not a well-known methodology, and thus it is even more important than usual to go above and beyond convincing the reader that this is a sound and reliable method. While including other papers as references to the method is great, a reader should not be required to track these down to make sense of the paper. Additionally, the figures that accompany the methods are non-intuitive and need better description– once again, making the paper, including the results section, quite difficult to follow.**

We are pleased to receive support from the reviewer. Obviously, as pointed out also by the other reviewers, we have to work on the accessibility of the

methods and results. We hope to have accomplished the accessibility by the restructured introduction and the new method subsection.

1. **Minor comments: Line 28: Change from 'they allow to infer' to 'they allow one to infer' Line 88: MCI has been previously defined Line 100-102 Remove the word 'well' Line 143- Add 'The' before following Line 164- Insert 'as' into 'Are not as much' Table A1 This table would be better with more information. Potentially adding ecosystem type/climate zone/type determined by PCMCI analysis of each fluxtower site would be useful.**

   Thanks for pointing to these potential improvements. We incorporated them in the manuscript.

2. **L114-What characteristics impinge on performance? This isn't explained. Also, why use precipitation as binary (rain/no rain) rather than a time series? Why can't precipitation be used in the model? Table B1 Shouldn't radiation be added as a variable?**

   The characteristics that impinge on network estimation (not performance) are those that are mentioned in the subsequent sentences. The difficulties of SWC will be added (see reviewer comment 1). We did try to incorporate precipitation as a time series and the time series has some binary character because it is typically zero (no precipitation) and seldom unequal to zero (precipitation). Such behaviour is a strong deviation from a normal distribution, needed for our independence test. Further it can happen that there is no precipitation during the estimation of a network. This however causes PCMCI to yield an error when attempting to standardise the time series.

   Radiation is incorporated in the network structure. Excluding Rg at the parameter 'selected_variables', Rg is set as the main driver and no effects of other variables on Rg are estimated (see answer to comment 11 of reviewer 1 and to comment 7 of reviewer 2).

**References**

[1] W. T. Crow, C. A. Gomez, J. M. Sabater, T. Holmes, C. R. Hain, F. Lei, J. Dong, J. G. Alfieri, and M. C. Anderson. Soil moisture–evapotranspiration overcoupling and l-band brightness temperature assimilation: Sources and forecast implications. *Journal of Hydrometeorology*, 21(10):2359–2374, 2020.

[2] P. A. Dirmeyer, F. J. Zeng, A. Ducharne, J. C. Morrill, and R. D. Koster. The sensitivity of surface fluxes to soil water content in three land surface schemes. *Journal of Hydrometeorology*, 1(2):121–134, 2000.

[3] D. J. Short Gianotti, A. J. Rigden, G. D. Salvucci, and D. Entekhabi. Satellite and station observations demonstrate water availability's effect on continental-scale evaporative and photosynthetic land surface dynamics. *Water Resources Research*, 55(1):540–554, 2019.

[4] I. N. Williams, Y. Lu, L. M. Kueppers, W. J. Riley, S. C. Biraud, J. E. Bagley, and M. S. Torn. Land-atmosphere coupling and climate prediction

over the us southern great plains. *Journal of Geophysical Research: Atmospheres*, 121(20):12–125, 2016.

---

## Referee Report (RR1)

BG-2020-374

**Summary**

The manuscript, "Functional convergence of biosphere-atmosphere interactions in response to meteorological conditions," investigates the causal and correlational interactions between of a number of variables measured across climate and ecosystem gradients at eddy covariance towers in the Fluxnet network. The primary unit of analysis in the manuscript is a multi-dimensional linked network of interaction strengths, as has been previously developed by the authors in an earlier study.

I was a previous reviewer for this manuscript and find the revisions to have significantly improved the presentation of methods, concepts, and analyses. I support publication of the manuscript at this time, either as is, or with minor clarifying revisions (see Specific Comments).

**General Comments**

I had a number of previous comments which I feel the authors have sufficiently addressed.

1) Clarification of the analytical framework:
   The introduction of the new Figure 1 greatly improves readability in my mind, and provides the necessary framing of concepts for readers to follow along with the authors in their exploration of concepts.
2) Theoretical assumptions:
   The revised manuscript contains explicit discussion of assumptions of PCMCI and when those assumptions may not be met in section 2.3 as well as explicit limitations of the study design in the new section 3.6.
3) Motivation of the causal analysis framework:
   The revised manuscript lays out a much clearer case for the causal framework used.
4) Further description of the distance correlations:
   The inclusion of section 2.5 defining the distance correlations helps in explaining the network space.
5) Overlapping data analysis windows:
   The previous manuscript emphasized the spatial coherence of the network points, which was problematic due to the fact that points shown in the 2d projected space all shared 2/3 of their data with other neighboring points. The revised manuscript de-emphasizes that discussion, removing the problem.

Even more generally, I feel like the addition of Figure 1 and the broader discussion of the archetypal networks greatly improves this manuscript. I think the study has a lot to offer readers, and that the insights included here are much more accessible following these revisions.

**Specific Comments**

1) I didn't fully follow the last part of the distance correlation definition in line 178. Is $|Xk-Xl|p$ the p-dimensional Euclidian distance between two p-dimensional vectors?

2) I might have missed it, but do you state what the numbers are in the upper left corners of the panels in Fig 3? Might be worth putting in the caption.

3) Exclusion of downwelling shortwave feedbacks:

   I think all of the reviewers were interested in what happens when Rg feedbacks are included in the analyses, as land-atmosphere feedbacks are at the core of the research questions asked by this manuscript. Their exclusion stands out in the methods, and I would have liked to see an analysis response somewhere in the response to reviewers, in supplementary discussion, or even in some summary statements in the methods or discussion (even if along the lines of, "Adding two-way linkages with Rg leads to a less interpretable 2d space – we believe this is because of violations of the stationary assumptions of the PCMCI method…").

   The comment, "We acknowledge the possibility of Rg being influenced by other variables… we presume this effect to be rather small" does not seem to justify omitting a variable from a multivariate analysis – you would check the effect size and quantitatively assess whether this is true (the same as the notion of a direct effect of NEE on VPD, which has less direct physical linkage than LE-->Rg).

   The new comment in line 126: "Besides these possibilities, setting Rg as driver can account for remaining non stationarities," gives some important motivation, and I can understand that if you want to load all of your non-stationarity somewhere, downwelling longwave seems like a good place to do it. At the same time, all of these data have been deseasoned, and so I wouldn't actually expect Rg to act any more exogenously than any of the other variables.

   I'm not sure whether this point will significantly impact a reader's experience, but it does seem to jump out a little to me, both before and after the manuscript's revisions.